# Compositional variability of Mg/Ca, Sr/Ca, and Na/Ca in the deep-sea bivalve *Acesta excavata* (Fabricius, 1779)

Nicolai Schleinkofer[1,2]*, Jacek Raddatz[1,2], David Evans[1,2], Axel Gerdes[1,2], Sascha Flögel[3], Silke Voigt[1,2], Janina Vanessa Büscher[4], Max Wisshak[5]

1 Institute of Geosciences, Goethe University Frankfurt, Frankfurt am Main, Germany, 2 Frankfurt Isotope and Element Research Center (FIERCE), Goethe University Frankfurt, Frankfurt am Main, Germany, 3 GEOMAR Helmholtz-Centre for Ocean Research, Kiel, Germany, 4 Department of Earth and Ocean Sciences, National University of Ireland, Galway, Ireland, 5 Marine Research Department, Senckenberg am Meer, Wilhelmshaven, Germany

* schleinkofer@em.uni-frankfurt.de

**Data Availability Statement:** All relevant data are within the manuscript and its Supporting Information files, and uploaded to PANGAEA doi: https://doi.pangaea.de/10.1594/PANGAEA.930296.

## Abstract

*Acesta excavata* (Fabricius, 1779) is a slow growing bivalve from the Limidae family and is often found associated with cold-water coral reefs along the European continental margin. Here we present the compositional variability of frequently used proxy elemental ratios (Mg/Ca, Sr/Ca, Na/Ca) measured by laser-ablation mass spectrometry (LA-ICP-MS) and compare it to *in-situ* recorded instrumental seawater parameters such as temperature and salinity. Shell Mg/Ca measured in the fibrous calcitic shell section was overall not correlated with seawater temperature or salinity; however, some samples show significant correlations with temperature with a sensitivity that was found to be unusually high in comparison to other marine organisms. Mg/Ca and Sr/Ca measured in the fibrous calcitic shell section display significant negative correlations with the linear extension rate of the shell, which indicates strong vital effects in these bivalves. Multiple linear regression analysis indicates that up to 79% of elemental variability is explicable with temperature and salinity as independent predictor values. Yet, the overall results clearly show that the application of Element/Ca (E/Ca) ratios in these bivalves to reconstruct past changes in temperature and salinity is likely to be complicated due to strong vital effects and the effects of organic material embedded in the shell. Therefore, we suggest to apply additional techniques, such as clumped isotopes, in order to exactly determine and quantify the underlying vital effects and possibly account for these. We found differences in the chemical composition between the two calcitic shell layers that are possibly explainable through differences of the crystal morphology. Sr/Ca ratios also appear to be partly controlled by the amount of magnesium, because the small magnesium ions bend the crystal lattice which increases the space for strontium incorporation. Oxidative cleaning with $H_2O_2$ did not significantly change the Mg/Ca and Sr/Ca composition of the shell. Na/Ca ratios decreased after the oxidative cleaning, which is most likely a leaching effect and not caused by the removal of organic matter.

**Funding:** The authors have received funding from DFG (Deutsche Forschungs Gemeinschaft (https://www.dfg.de/)).The Project number is 383168872 (https://gepris.dfg.de/gepris/projekt/383168872). Recipient of funding is Dr. Jacek Raddatz (J.R.) The funders had no role in study design, data collection and analysis, decision to publish, or preparation of the manuscript.

**Competing interests:** The authors have declared that no competing interests exist.

## Introduction

Cold-water coral (CWC) reefs comprise an important contribution to the marine biodiversity on continental margins that is similar to that of warm-water coral (WWC) reefs (Shannon-index = CWC: 5.5, WWC: 5.09 [1–6]). The reefs provide shelter for other organisms such as the bivalve *Acesta excavata* (Fabricius, 1779) as one of the key species associated with CWC reefs. These sensitive ecosystems are greatly threatened by the combined effects of ocean acidification and warming [7–10]. Understanding the physiological limits of these ecosystems and their associated organisms in the past may provide important information for conservation strategies. Analyzing the environmental factors that control the distribution of CWCs has therefore been the focus of marine research in the last few decades. Salinity, temperature and aragonite saturation are important factors controlling the distribution of CWC [11, 12]. Water flow velocity and the effect of changing flow velocities on food supply to the corals also control CWC reef distribution [13], whilst the degree to which seasonality controls the flourishing state of CWC reefs is poorly constrained. Seasonally fluctuating zooplankton concentrations are an important nutrient source for the corals and might thus be an important controlling factor for the distribution of CWC reefs [11, 14]. Moreover, the synchronization of gamete spawning may also be triggered by seasonal changes in temperature [15] or changing moon phases [16]. However, the reconstruction of past changes in the seasonality at CWC reefs is challenging because reef-forming CWCs, such as *Desmophyllum pertusum* (formerly known as *Lophelia pertusa*) lack well-defined growth patterns. While growth patterns in reef forming CWCs are observable, the timing of their formation is unknown [17, 18], making seasonal reconstructions challenging. In addition, commonly used proxies such as Mg/Ca and can be influenced by vital effects [19–22]. Other corals such as bamboo corals show more promising results regarding growth patterns and trace element proxies [23–26] and might therefore provide an alternative archive.

The deep-water bivalve *Acesta excavata* could be an archive for the seasonality of seawater attributes in regions of CWC distribution. A previous study revealed cyclic repetitions of density changes in the shell and regular growth increment spacing, indicating a rhythmic, possibly annual control of shell deposition [27]. In general, *A. excavata* displays two growth modes. The first phase lasts until the bivalve has built 18–22 increments (= 18–22 years [27]) and reaches a size of approximately 10 cm. The second growth phase is characterized by slower growth and more tightly aligned growth increments [27]. The growth increments suggest a typical lifespan of 50 to 80 years [27]. The change of growth mode happens simultaneously with a sex change from male to female [28]. *A. excavata* features a semi-continuous reproductive cycle with one spawning event in May/June and another one beginning in August and lasting the rest of the year [28]. The annually formed increments make these species a good candidate for paleoenvironmental reconstructions using proxies such as Element/Ca ratios.

The majority of studies on bivalves conclude that possible temperature controls on elemental ratios such as Mg/Ca and Sr/Ca are strongly modulated by vital-effects [29–36]. Similarly for Na/Ca, controlling factors for Na incorporation into marine biogenic carbonates are not fully understood but possible mechanisms include salinity (Foraminifera [37–39]), the Na/Ca ratio of the ambient water (Foraminifera [40, 41]), temperature (CWC [37, 42, 43]) and growth rate related effects due to lattice defects and distortions [44]. Marine clams, such as the aragonitic bivalve *Arctica islandica*, show no reproducible Na/Ca time-series in specimens from the same location [45], which makes external forcing mechanisms unlikely. Whether this assumption holds true for other species has to be further investigated.

Here, we present, element to calcium ratios measured with laser-ablation mass spectrometry (LA-ICP-MS) from *A. excavata* compared to an *in-situ* instrumental record. Live

specimens of *A. excavata* were collected from two Norwegian CWC reefs at 200–300 m to explore possible environmental and biological controls on Mg/Ca, Na/Ca, and Sr/Ca. Environmental data was gathered via two landers deployed in close proximity to the investigated reefs and provide one-year-long records of high-resolution data, which can be used for direct comparison. This data is a key part of this investigation, as *in-situ* environmental data from deep-waters is scarce and difficult to acquire. Resulting proxies from this study have the potential to improve our understanding of the physiological limits of CWC reef distribution in the past and could provide necessary information about the future of these important structures regarding the present-day climate change. This is especially important with regards to the location of our archive in intermediate water depths as there are not many archives from such depths. In addition, we test the effect of an oxidative cleaning step (using $H_2O_2$) on LA-ICP-MS trace element data.

## Material and methods

### Sampling

Eight specimens of *A. excavata* from the Norwegian Atlantic region were investigated in this study. Four of them were collected in the Sula Reef (N 64˚06.64'/E 08˚07.11', depth ~ 300 m) and four were collected in a reef close to the island Nord-Leksa at the entrance of the Trondheimsfjord (N 63˚36.47'/E 09˚23.03', depth ~ 200 m) (Fig 1). All specimens were collected live with the manned submersible JAGO from GEOMAR [46] during RV POSEIDON [47] cruise

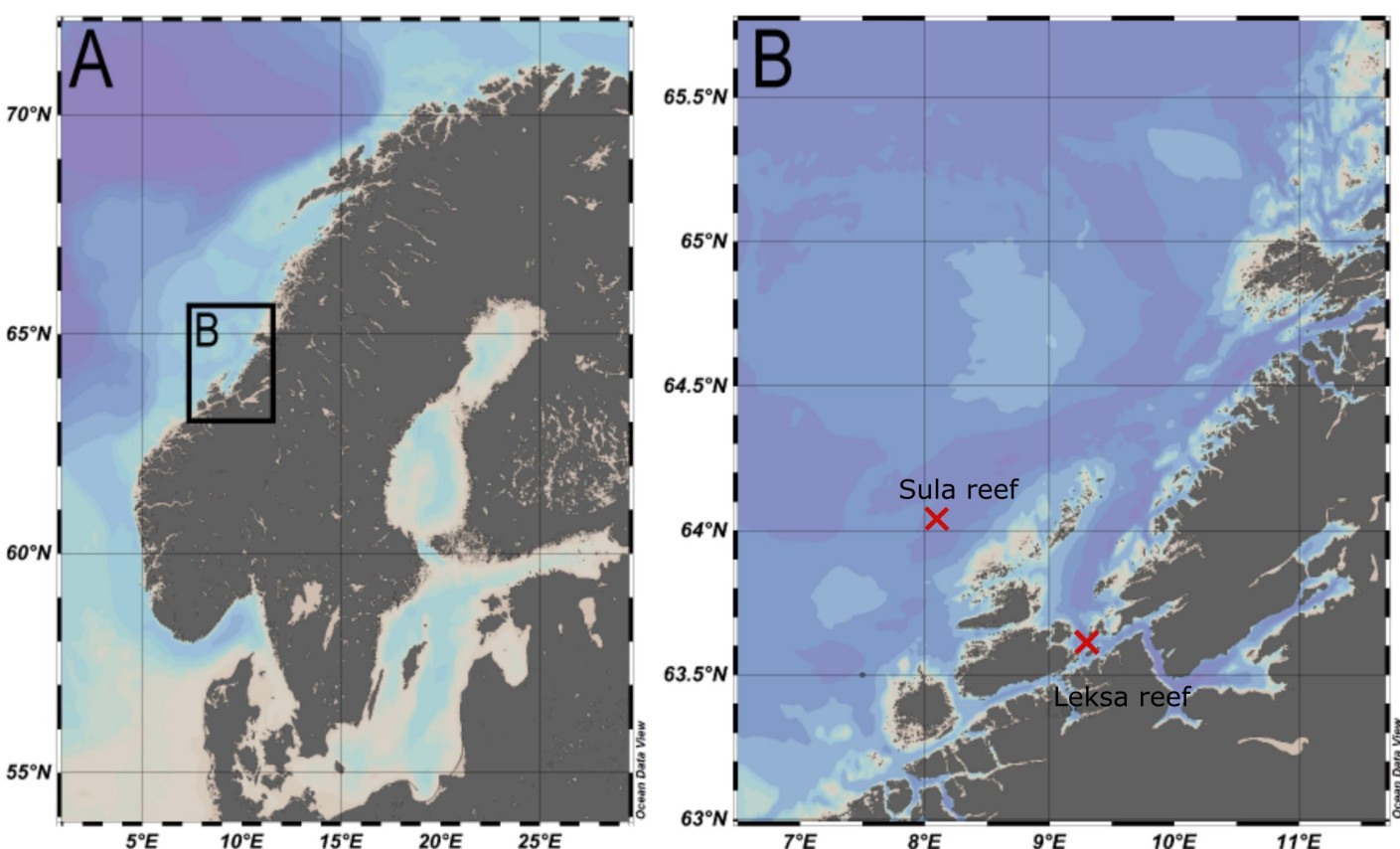

**Fig 1. Map of the sampling locations.** A. Overview of the Fennoscandian Peninsula. B. Enlarged section of the sampling locations.

473 [48]. Permission for entering and sampling in Norwegian waters was granted by the Director-ate of Fisheries and the Norwegian Armed Forces. Specific permissions to sample this particular species were not necessary as *A. excavata* does not appear in CITES lists and the Nagoya protocol was not yet established at the time of sampling. Directly after sampling the soft body was physi-cally removed and the shells were dried in an oven at 40˚C. Cleaning was conducted on board with knives as well as prior to further investigation in an ultrasonic bath (5 min).

Slabs of 8 mm thickness were then cut along the major growth axis of each shell and 20 mm-long sub-samples were taken of the ontogenetic oldest part of the bivalve (ventral side; Fig 2). These samples were mounted vertically into circular mounts and embedded in epoxy resin. Prior to the acquisition of trace element data using LA-ICP-MS, the slab surface was ground with 9 μm grid with silicon carbide sanding paper and then polished using 3 μm dia-mond-water based lapping paste.

Additional shell strips were prepared as thin sections and treated with Mutvei's solution (50 min, 45˚C) to enhance growth increment visibility [49]. Pictures of the investigated specimens were taken with a Keyence VHX-6000 digital microscope with 200X magnification. Size mea-surements were conducted with the Keyence software, whereas size measurements of the shells were conducted with digital calipers (Mitutoyo).

## Oceanographic data

Seawater properties (temperature, conductivity (Nord-Leksa Reef) and flow velocity (Sula Reef) were provided from a lander study conducted during research cruises (POS455 and POS473) with RV POSEIDON in the Norwegian Sea. The data used here is part of a separate study comprised of oceanographic data assimilation of multiple parameters over a full annual cycle, which is beyond the scope of the present study, but of which temperature in particular was used to correlate it to the shells from the same origin. Two Satellite Lander Modules (SLMs, GEOMAR) were deployed in the Leksa Reef. Each module was equipped with Seabird SBE16 PLUS CTD sensors (conductivity (±0.0005 S/m), temperature (±0.005˚C), depth) as well as WETLabs ECO-FLNTU(RT)D turbidity & chlorophyll sensors and Seabird SBE43 oxy-gen sensors. The lander at Sula Reef was an Aanderaa Seaguard RCM mounted in a pyramid-shaped POM frame equipped with a current meter (linear flow velocity and absolute direction) as well as CTD sensors. The landers were deployed on July 2nd, 2013 (Trondheimsfjord) at a depth of 215m and July 4th, 2013 (Sula Reef) at a depth of 315 m. Data in the Sula Reef was recorded every 30 minutes, in the Nord-Leksa Reef every 15 minutes resulting in 20123 and 37603 data points, respectively. This translates to a total recorded time of 419 days in the Sula Reef and 391 day in the Nord-Leksa Reef [48].

For the Sula Reef, temperature data from the ARGO project [50] were used for additional comparison. The data utilized here consists of composite temperature measurements from 5–9˚E/63.5–64.5˚N, as such temperatures from locations to the northeast of the CWC study sites are also included.

## Elemental analysis by LA-ICP-MS

The polished mounts to be used for *in-situ* elemental analysis were cleaned in an ultrasonic bath with ethanol. Two samples (6R & 14R) were measured twice, prior to and after submerging them for 1 hour in an alkaline 5% $H_2O_2$ solution [51]. Elemental compositions were measured in the outer calcitic shell portion (fibrous material) (Fig 2B), 0.1 mm below the shell surface and in the microgranular calcite layer parallel to the fibrous section [27]. Laser ablation was per-formed using a Resolution M50 193 nm ArF Excimer Laser system (Resonetics), with a 72 μm beam diameter, a pulse rate of 10 Hz and 10 μm/s scan speed. Total sweep time was 0.65 s. Prior

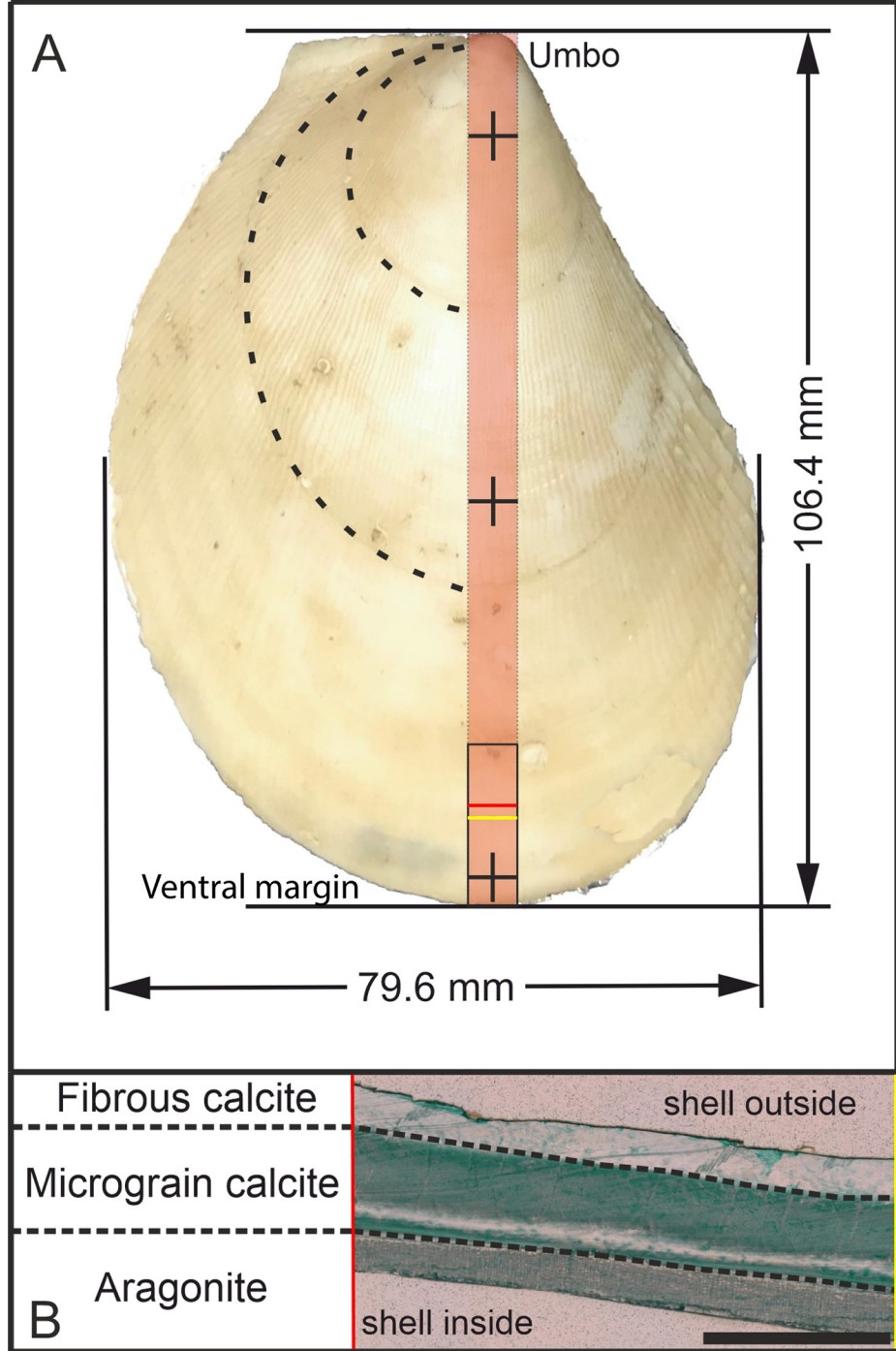

**Fig 2. Sample overview and crystalline phases in the shell of *A. excavata*.** A: Sample 17R from the Sula Reef. The red area marks the cut slab and the black rectangle shows the investigated area. Crosses mark the position of shell thickness measurements (15 mm from umbo, 50 mm from umbo and 5 mm from ventral margin). The shell length was measured perpendicular to the bivalve auricle and the width was measured parallel to the auricle of the bivalve along the maximum distance. Dashed lines show visible growth lines B: Cross-section of the shell (location marked in A with red and yellow lines), colored with Mutvei's solution [49]. Scalebar is 1 mm.

to the measurement a fast precleaning pass was conducted at 0.2 mm/s, 10Hz, and 104 μm laser spot size. Elemental ratio analysis was performed with a Thermo-Scientific ELEMENT XR sector field ICP-MS. NIST SRM 612 glass was used as the external standard and the MACS-3 nano-pellet standard was used to assess accuracy and precision. We used 62.4 μg/g for the NIST 612 Mg concentration [52], 78.4 μg/g for Sr and 101000 μg/g for Na [53]. Standards were ablated in an identical manner to the samples. The monitored isotopes (m/z) were $^{23}$Na, $^{24}$Mg, and $^{88}$Sr. $^{43}$Ca was used as the internal standard and for E/Ca calculation. Data processing was conducted in Excel without further specialized software. Accuracy and precision, assessed via repeat measurements of the MACS-3 standard (n = 5) resulted in a measured Mg/Ca ratio of 8.2 ± 0.3 mmol/mol (reference value = 7.8 ±0.4 mmol/mol [54]), Na/Ca: 25.7 ± 1.3 mmol/mol (reference value = 27.3 ± 1.8 [54]) and Sr/Ca: 8.5 ± 0.3 mmol/mol (reference value = 8.2 ± 0.4 mmol/mol [54]). We additionally used NIST SRM 610 as an alternative external standard but the results show no significant differences in the Mg/Ca ratios compared to using NIST SRM 612 as the primary standard (when using 62.4 μg/g as the Mg concentration [52]).

## Organic content and fluorescence microscopy

The organic content of the shell was measured by combustion analysis. Around 45 mg of powdered sample material was ground from the calcitic and the aragonitic shell section. The sample powder was heated in a furnace for 1 hour at 105°C to remove the water and was thereafter heated to 500°C for 20 hours to combust organic matter. The samples were weighed after every step to determine water and organic content. Marble and quartz powder standards were monitored to ensure reliable results.

In addition, we used fluorescence microscopy to investigate the distribution of the organic material in the shell. Fluorescent images were taken using a Leica DMRX-POL microscope with fluorescent front light and a 50W mercury lamp. The microscope is equipped with an H3 and D filter cube, which excites in the wavelength range of blue—violet (Bandpass filter: 420–490 nm) and violet—ultraviolet (Bandpass filter: 355–425 nm) [55]. The pictures were taken with a digital camera connected to the microscope with ¼ - 10 s exposure time.

## Statistical analysis

All statistical calculations were conducted with OriginPro 2020. We used the T-test to compare means of different populations and a linear regression model to investigate the relationship between predictor and response variables. In addition, we used a multiple linear regression model:

$$y_i = \beta_0 + \beta_1 x_1 + \beta_2 x_2 + \cdots + \beta_n x_n + \varepsilon_n \tag{1}$$

to predict response variables from multiple predictor (temperature and salinity) variables.

## Results

### Recorded temperature and salinity data

The recorded environmental data shows a clear annual variability (Fig 3). Temperature varies between 7.7°C in the summer months and 8.5°C in the winter months in the Leksa Reef. High temperature and salinity in winter are caused by the replacement of the more brackish and colder Norwegian Coastal Current (NCC) by the warmer, saltier North Atlantic Current [56, 57]. This is an effect of increased meltwater supply and increasing northerly winds, which causes the depth of the NCC to decrease. The shallower NCC then allows the NAC water to replace the bottom water in the fjord. The temperature in the Sula Reef varies between 7°C

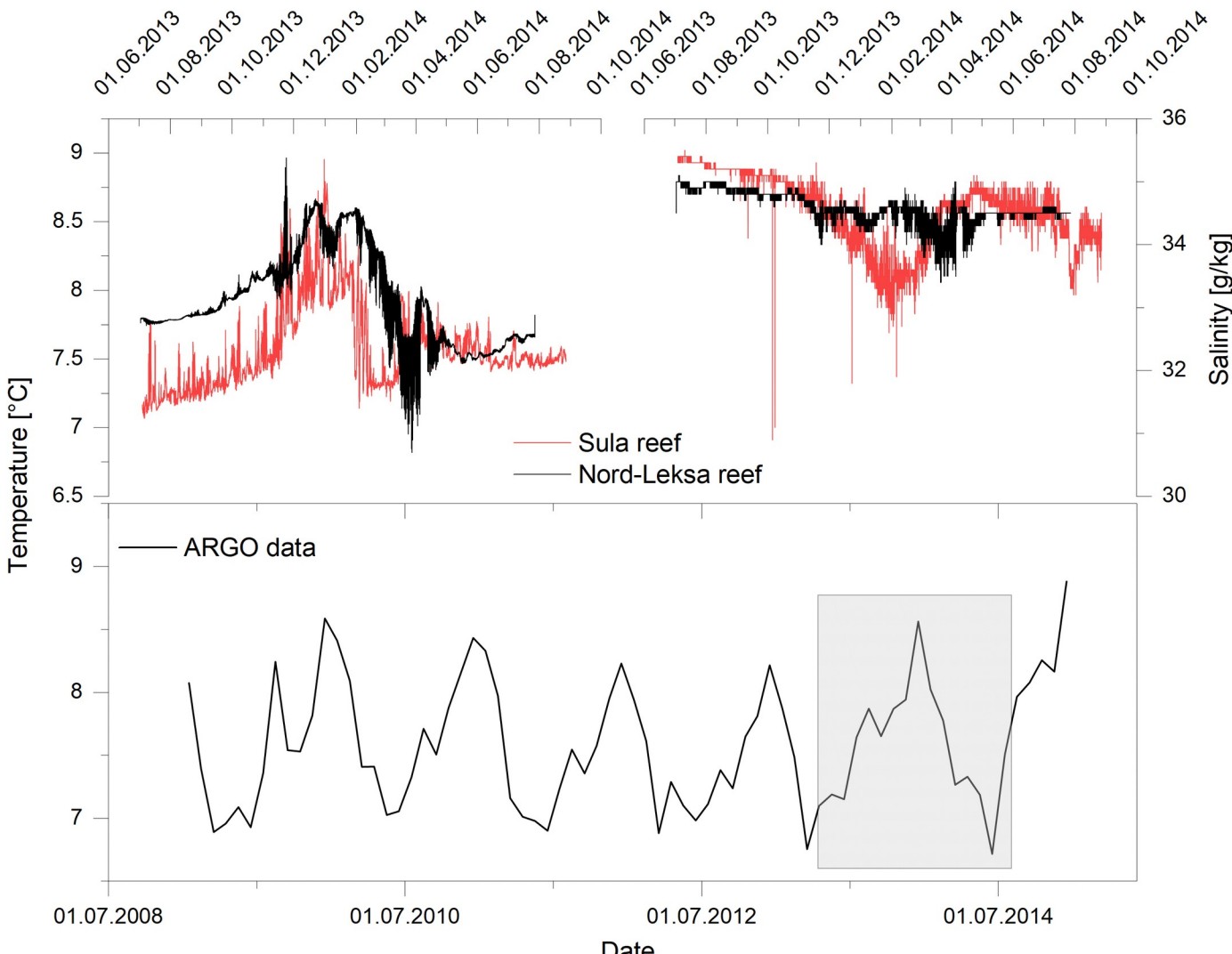

**Fig 3. Environmental data gathered by two landers and ARGO data.** A) Temperature. B) Salinity. C) ARGO Temperature data. The grey box gives the timeframe of the lander deployments.

and 8.8˚C. Seawater salinity is highest during winter and low in summer, varying from 33.5–36 g/kg in the Sula Reef and 34–36 g/kg in the Leksa Reef.

## Microscopic shell properties

The uppermost fibrous zone is similar to the underlying microgranular zone in its appearance. Following treatment with Mutvei's solution, the distinct identification is possible, since the fibrous zone is weakly stained compared to the microgranular zone (Fig 2). The aragonitic zone shows a striped pattern consisting of gray and white bands. The relative thickness of the different shell portions is similar between investigated specimens, although variations within specimens are visible. Within the microgranular section a white band is visible that lies about 100 µm above the aragonitic zone. This white band appears to be related to the aragonitic zone as it runs parallel and is discontinued in the youngest shell portions where the aragonitic section disappears. Growth lines are faintly visible, however, a differentiation between yearly

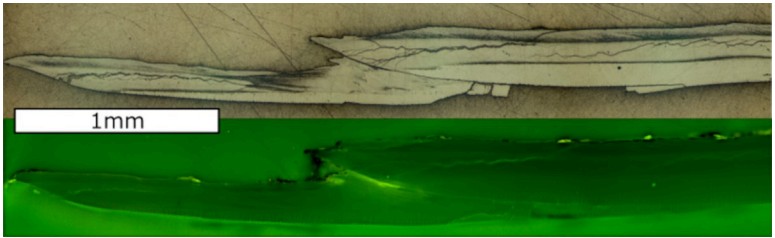

**Fig 4. Microscopic shell images and elemental ratios of *A. excavata*.** A) Mg/Ca, Sr/Ca and Na/Ca ratios of the fibrous (black line) and microgranular (red line) shell section of sample 14R plotted against distance from the ventral margin. B) Red lines indicate assumed yearly growth lines as seen from microscope pictures of the shell. C) Enlarged section of B) with the jagged edge on the shell surface (red circle) and dark areas (red square) from where the growth line emerges. Yellow dashed lines mark the laser track in the fibrous and microgranular shell layer. Black arrows in D mark microgrowth lines. Scale bars are 1 mm long. Width of the picture in panel D is 1mm. Additional figures are given in the (S7 Appendix).

growth increments and microgrowth increments cannot be made as there is no difference in size, thickness or other morphological features. The growth lines span both the fibrous and the microgranular zone but are not present in the aragonitic zone. Some growth lines start from a dark colored area at the aragonite—calcite transition [27] and end in a jagged edge on the shell surface (Fig 4). These lines coincide with minima in Mg/Ca ratios and growth lines visible on the outer shell surface and are therefore regarded as annual growth lines [58]. The microgrowth increments are hard to count due to their poor visibility, but where counting was possible, samples show 20 to 30 micro growth increments between two major growth lines.

## Organic content

The combustion experiments revealed low concentrations of organic material in the shell of *A. excavata* (Table 1). An excitation of autofluorescence in growth lines was not possible with either of the two filter cubes and wavelengths. The fluorescent microscope images show that the organic matter is not concentrated in specific areas (Fig 5).

## Elemental composition of different shell layers

**Fibrous shell section.** Measured Mg/Ca ratios of the eight investigated bivalves vary between 7.9 and 48.3 mmol/mol with a mean of 16.4 mmol/mol (median = 14.97 mmol/mol) and 1 standard deviation (SD) of 5.5 mmol/mol for all investigated specimens. Samples from the Trondheimsfjord show higher mean values of 17.8 mmol/mol (median = 16.35 mmol/mol, min = 9.7 mmol/mol, max = 48.3 mmol/mol) compared to 15.1 mmol/mol (median = 13.65 mmol/mol, min = 7.9 mmol/mol, max = 40.2 mmol/mol) in the Sula Reef. Every investigated bivalve shows a well-developed pattern of minimum-maximum variations with relatively stable baseline values around 10 mmol/mol punctuated by repeating sharp peaks up to 48 mmol/mol (Fig 4 and S7 Appendix).

**Table 1. Results of combustion experiments.**

|  | Weightloss after A (water content) | Weightloss after B (organic content) |
|---|---|---|
| Aragonite | 0.44% | 1.81% |
| Calcite | 0.53% | 1.50% |
| STD Marble | 1.15% | 0.40% |
| STD Quartz | 0.40% | 0.05% |

A = 105˚C for 1h, B = 500˚C for 20h.

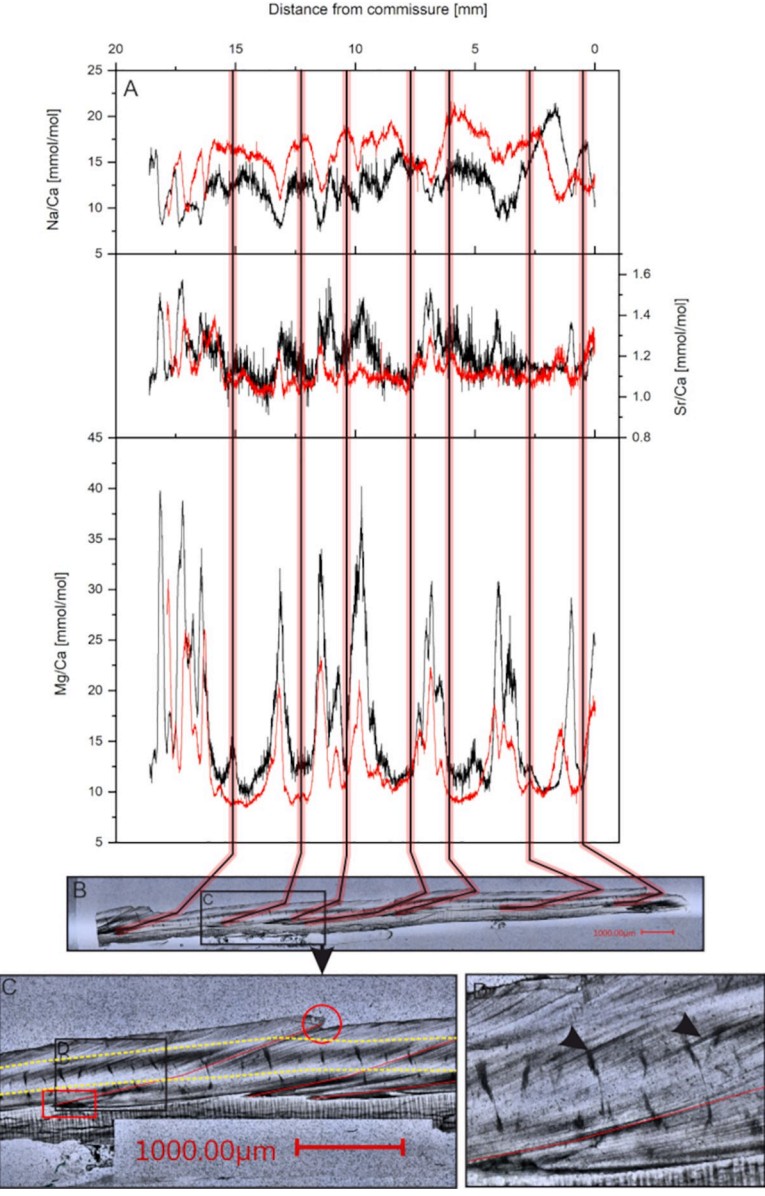

**Fig 5. Light microscope and fluorescent microscope images of sample 6R.** Magnification = 25X, H3 filtercube, ¼ s exposure time. Scale bar is 1 mm.

Sr/Ca ratios vary between 0.8 and 2.0 mmol/mol with a mean of 1.2 mmol/mol and 1SD of 0.1 mmol/mol. Sr/Ca ratios are only show slight variations of the measured distance.

Na/Ca ratios vary between 7.4 mmol/mol and 29.3 mmol/mol with a mean of 16.8 mmol/mol and 1SD of 3.8 mmol/mol. Similar to Mg/Ca, Na/Ca ratios show a distinct pattern of reoccurring minima and maxima.

The effects of treating the samples with $H_2O_2$ were different for Mg/Ca, Na/Ca and Sr/Ca. Measured Mg/Ca ratios were not significantly different between treated and untreated specimens (Table 2) whereas Sr/Ca ratios displayed significant differences after the $H_2O_2$ treatment in the Sula Reef sample but not in the Nord-Leksa sample. Na/Ca ratios decreased significantly by 2.53 and 3.17 mmol/mol after the $H_2O_2$ treatment.

**Table 2. Mean E/Ca ratios (mmol/mol) and differences before and after H₂O₂ treatment.**

| Sample | before H₂O₂ | | | after H₂O₂ | | | Difference | | |
|---|---|---|---|---|---|---|---|---|---|
| | Mg/Ca | Na/Ca | Sr/Ca | Mg/Ca | Na/Ca | Sr/Ca | Mg/Ca | Na/Ca | Sr/Ca |
| 6R | 16.67 | 16.42 | 1.25 | 16.77 | 13.89 | 1.24 | +0.1 | -2.53 | -0.01 |
| 14R | 16.53 | 16.04 | 1.3 | 16.73 | 12.87 | 1.2 | +0.2 | -3.17 | -0.1 |
| T-test results | | | | | | | | | |
| Sample | DF | Mg/Ca | | | Na/Ca | | | Sr/Ca | |
| | | t | p | | t | p | | t | p |
| 6R | 3875 | -1.17 | 0.24 | | -32.08 | 1.6E-200 | | 1.07 | 0.28 |
| 14R | 3983 | -0.5 | 0.62 | | 38.95 | 1.8E-281 | | -25.9 | 3.9E-137 |

Differences between the two treatments were tested for significance with a T-test. E/Ca ratios are reported in mmol/mol. DF = Degrees of Freedom.

**Micrograndular shell section.** Within the micrograndular shell section the overall trend of the curves is similar to those from the fibrous sections (Fig 4 and S7 Appendix), although the absolute values are different. Significant mean differences are observable for every E/Ca ratio in all measured samples (Table 3). Mean Mg/Ca ratios decrease by 2% to 39% and Na/Ca ratios increase in the Micrograndular shell layer by 5% to 34%. Mg/Ca ratios are more influenced in the peaks while baseline values remain unchanged. This is different for Na/Ca ratios, which show alterations in both minimum and maximum values. Unlike Mg/Ca and Na/Ca, Sr/Ca ratios display no systematic difference between the two investigated layers. Three samples show lower and five samples show higher mean Sr/Ca ratios in the micrograndular shell layer compared to the fibrous shell layer.

**Table 3. Mean E/Ca ratios (mmol/mol) and differences between the fibrous and micrograndular shell layer.**

| Sample | Fibrous | | | Micrograndular | | | Difference | | |
|---|---|---|---|---|---|---|---|---|---|
| | Mg/Ca | Na/Ca | Sr/Ca | Mg/Ca | Na/Ca | Sr/Ca | Mg/Ca | Na/Ca | Sr/Ca |
| 1R | 17.47 | 17.87 | 1.2 | 12.72 | 21.18 | 1.12 | +4.75 | -3.31 | +0.08 |
| 6R | 16.73 | 13.89 | 1.24 | 13.56 | 16.38 | 1.28 | +3.17 | -2.49 | -0.04 |
| 11R | 16.64 | 20.47 | 1.07 | 11.69 | 20.21 | 1.02 | +4.95 | +0.26 | +0.05 |
| 12R | 20.18 | 13.56 | 1.23 | 12.2 | 17.27 | 1.3 | +7.98 | -0.71 | -0.07 |
| 14R | 16.77 | 12.87 | 1.2 | 12.86 | 15.67 | 1.13 | +3.91 | -2.8 | +0.07 |
| 16R | 16.54 | 18.26 | 1.18 | 14.6 | 19.71 | 1.23 | +1.94 | -1.45 | -0.05 |
| 17R | 12.45 | 17.56 | 1.14 | 10.33 | 23.64 | 1.18 | +2.12 | -6.08 | -0.04 |
| 25R | 14.59 | 19.85 | 1.11 | 9.9 | 23.12 | 1.16 | +4.69 | -3.27 | -0.05 |
| T-test results | | | | | | | | | |
| Sample | DF | Mg/Ca | | | Na/Ca | | | Sr/Ca | |
| | | t | p | | t | p | | t | p |
| 1R | 4998 | -38.4 | 2.3E-283 | | -35.9 | 5.1E-251 | | 28 | 1.4E-160 |
| 6R | 5040 | 24.3 | 3.3E-123 | | -36.5 | 3.6E-259 | | -11 | 2.8E-30 |
| 11R | 5455 | 40 | 2.4E-286 | | 3.8 | 1.6E-4 | | 16.7 | 4.4E-61 |
| 12R | 5434 | 37.6 | 2.0E-275 | | -59.7 | 2.4E-249 | | -33.1 | 1.1E-218 |
| 14R | 5102 | 25.8 | 4.4E-138 | | -40.7 | 2.6E-234 | | 27.1 | 3.1E-151 |
| 16R | 5531 | 13.5 | 1.1E-40 | | -14.4 | 2.6E-46 | | -20.3 | 1.9E-88 |
| 17R | 5192 | 29.0 | 1.1E-171 | | -86.4 | 3.5E-234 | | -23.3 | 1.3E-114 |
| 25R | 5332 | 87.9 | 3.3E-253 | | -66.0 | 3.1E-263 | | -26.43 | 9.2E-145 |

Differences between the two shell layers were tested for significance with a T-test. E/Ca ratios are reported in mmol/mol. DF = Degrees of Freedom.

**Table 4. Coefficients of determination r$^2$, slope and *p*-values of temperature and salinity with the investigated elemental ratios.**

| Sample | | Mg/Ca | Na/Ca | Sr/Ca | Mg/Ca | Na/Ca | Sr/Ca |
|---|---|---|---|---|---|---|---|
| | DF | Water temperature | | | Salinity | | |
| 1R | 596 | r$^2$ = 0.01 | 0.14 | 0.1 | 0.05 | <0.01 | 0.24 |
| | | Slope = 1.8 | -3.4 | -0.09 | -3.2 | -0.09 | -0.1 |
| | | *p* = 0.001 | <0.001 | <0.001 | <0.001 | 0.8 | <0.001 |
| 11R | 326 | 0.55 | 0.04 | 0.39 | 0.13 | 0.03 | 0.15 |
| | | 4.9 | -1.2 | -0.07 | 2.1 | 0.9 | -0.04 |
| | | <0.001 | <0.001 | <0.001 | <0.001 | <0.001 | <0.001 |
| 12R | 189 | 0.55 | 0.11 | 0.54 | 0.03 | 0.54 | 0.34 |
| | | -10.2 | -1.2 | -0.28 | -2.4 | -2.6 | -0.2<0.001 |
| | | <0.001 | <0.001 | <0.001 | 0.01 | <0.001 | <0.001 |
| 14R | 329 | <0.001 | 0.35 | 0.01 | 0.59 | 0.29 | 0.31 |
| | | -0.11 | -5.7 | -0.03 | -8.9 | 3.25 | -0.07 |
| | | 0.9 | <0.001 | 0.03 | <0.001 | <0.001 | <0.001 |
| 16R | 259 | 0.63 | 0.57 | 0.17 | 0.01 | 0.12 | <0.01 |
| | | 10.66 | -6.2 | 0.09 | -1.2 | 2.35 | 0.01 |
| | | <0.001 | <0.001 | <0.001 | 0.06 | <0.001 | 0.38 |
| 17R | 503 | 0.50 | 0.11 | 0.02 | 0.15 | 0.01 | 0.06 |
| | | 11.2 | -2.0 | 0.04 | -3.7 | -0.4 | -0.04 |
| | | <0.001 | <0.001 | <0.001 | <0.001 | 0.02 | <0.001 |
| 25R | 295 | 0.05 | 0.02 | <0.01 | 0.01 | 0.11 | <0.01 |
| | | 1.4 | -0.9 | <0.01 | -0.5 | 1.6 | <0.01 |
| | | <0.001 | 0.02 | 0.85 | 0.08 | <0.001 | 0.4 |
| All Samples | 2509 | 0.06 | 0.02 | 0.03 | 0.02 | 0.04 | 0.06 |
| | | 3.8 | -1.1 | -0.05 | -1.6 | 1.5 | -0.06 |
| | | <0.001 | <0.001 | <0.001 | <0.001 | <0.001 | <0.001 |

Correlations between E/Ca and temperature-salinity are calculated with the same interval for each sample. The correlation for all samples is a combined regression with every sample using the same time interval. Sample 6R could not be tested as there is a particle embedded in the shell in the specific area. DF = Degrees of Freedom.

## Correlating element/Ca data to instrumental data

The temperature data from the deployed landers were compared to the measured Mg/Ca data to explore the relationship between the parameters. Based on previous studies [27] and our own observations, we assume the growth lines were produced on an annual basis. We therefore identified the last minimum to minimum Mg/Ca cycle based on occurring growth lines, i.e. the cycle closest to the ventral margin of the shell, to which we then compared the recorded temperature. This reveals no overall correlation between Mg/Ca and seawater temperature (r$^2$ = 0.06, *p*<0.05) (Table 4). Four of the investigated samples (11R,12R,16R,17R) show high correlation coefficients (r$^2$ = 0.5–0.63) whereas other samples show poor or insignificant correlation. Similarly, Sr/Ca and Na/Ca ratios show no significant correlations with water temperature. Again, some samples show coefficients of determination of up to r$^2$ = 0.57.

Multiple linear regression with temperature and salinity as predictor variables and element/Ca as dependent variables show that temperature and salinity can account for a moderate to high amount of variability (33–79%) in Mg/Ca ratios (Table 5). This is also true for Sr/Ca and Na/Ca ratios. Sample 25R from the Sula reef distinguishes from the others by not displaying significant correlations for Mg/Ca and Sr/Ca ratios.

**Table 5. Coefficients of determination $r^2$ of multiple linear regressions with temperature and salinity as predictor variables and element/Ca as dependent variables.**

|  | DF | Mg/Ca | T | S | Na/Ca | T | S | Sr/Ca | T | S |
|---|---|---|---|---|---|---|---|---|---|---|
| 1R | 595 | 0.33 | Sl = 16.9 | -15.7 | 0.46 | -11.4 | 8.3 | 0.28 | 0.1 | -0.2 |
|  |  |  | $p < 0.001$ | <0.001 |  | <0.001 | <0.001 |  | <0.001 | <0.001 |
| 11R | 325 | 0.79 | 9.97 | -5.2 | 0.47 | -7.3 | 6.3 | 0.45 | -0.1 | 0.04 |
|  |  |  | <0.001 | <0.001 |  | <0.001 | <0.001 |  | <0.001 | <0.001 |
| 12R | 188 | 0.61 | -12.37 | 3.9 | 0.54 | 0.3 | -2.7 | 0.59 | -0.2 | -0.1 |
|  |  |  | <0.001 | <0.001 |  | 0.15 | <0.001 |  | <0.001 | <0.001 |
| 14R | 328 | 0.58 | 0.24 | -8.9 | 0.66 | -5.9 | 3.3 | 0.32 | -0.02 | -0.07 |
|  |  |  | 0.71 | <0.001 |  | <0.001 | <0.001 |  | 0.03 | <0.001 |
| 16R | 258 | 0.64 | 10.8 | 0.6 | 0.61 | -5.8 | 1.4 | 0.19 | 0.1 | 0.03 |
|  |  |  | <0.001 | 0.12 |  | <0.001 | <0.001 |  | <0.001 | 0.008 |
| 17R | 502 | 0.59 | 10.5 | -2.9 | 0.13 | -2.2 | -0.6 | 0.07 | 0.03 | -0.04 |
|  |  |  | <0.001 | <0.001 |  | <0.001 | <0.001 |  | 0.02 | <0.001 |
| 25R | 294 | 0.04 | 1.6 | 0.15 | 0.11 | 0.6 | 1.8 | 0.002 | 0.005 | 0.009 |
|  |  |  | <0.001 | 0.66 |  | 0.2 | <0.001 |  | 0.74 | 0.38 |

Slope = Sl and *p*-value of the regression are presented in column T = temperature and S = salinity. Sample 6R could not be tested as there is a particle embedded in the shell in the specific area. DF = Degrees of Freedom.

## Shell linear extension rate

Shell extension rates were calculated by measuring the distance between two maximum Mg/Ca peaks. Annual extension rates of the investigated shells are reported in Table 6. Except for two specimens, all investigated samples show similar annual extension rates between 1 mm/a and 4.8 mm/a. Only the samples 17R and 25R show higher annual rates between 6.9 mm/a and 7.9 mm/a. Both these samples were collected in the Sula Reef. We observe a significant inverse correlation between both Mg/Ca (DF = 12, $r^2$ = 0.63, *p*<0.001) and Sr/Ca (DF = 12, $r^2$ = 0.38, *p* = 0.02) with linear extension rate. No significant correlation is found between Na/Ca and linear extension rate (DF = 12, $r^2$ = 0.09, *p* = 0.57).

## Discussion

### Sclerochronology

Sclerolomorphological features are difficult to interpret in *A. excavata* due to their poor visibility (weak contrast). The main growth lines described earlier, which emerge from dark areas

**Table 6. Linear extension rates.**

| Sample | Linear extension rate [mm/a] | | | | | | | | | | |
|---|---|---|---|---|---|---|---|---|---|---|---|
|  | 2004 | 2005 | 2006 | 2007 | 2008 | 2009 | 2010 | 2011 | 2012 | 2013 | 2014 |
| 1R |  |  | 2.1 | 2.1 | 2.1 | 2.1 | 1.2 | 0.8 | 1.1 | 4.1 |  |
| 6R |  |  |  |  | 1.4 | 3.8 | 4.8 | 2.5 |  |  |  |
| 11R |  |  |  |  | 3.4 | 3.6 | 3.9 | 2.9 | 1.5 | 2.4 |  |
| 12R | 2.4 | 2.4 | 2.3 | 2.3 | 1.0 | 1.0 | 1.2 | 1.6 | 2.3 | 1.4 |  |
| 14R |  |  |  | 1.7 | 1.8 | 1.8 | 1.8 | 2.9 | 2.9 | 1.8 |  |
| 16R |  |  | 1.6 | 1.6 | 1.6 | 2.4 | 1.4 | 3.4 | 4.8 | 2.4 |  |
| 17R |  |  |  |  |  |  |  |  | 7.9 | 7.4 |  |
| 25R |  |  |  |  |  |  | 7.2 | 6.9 | 3.9 |  |  |

Extension rates for Sample 6R are missing for the years 2012 and 2013 due to embedded particles in the shell. Extension rate for the year 2104 cannot be reported since the samples were collected in summer 2014.

and end in external shell features are interpreted as yearly growth increments (Fig 4) due to the cyclic E/Ca variations within these growth increments and based on results of previous studies [27]. The dark areas could be caused by high amounts of organic matter (organic matrices) that are embedded in the shell during times of low calcification rates [59]. However, the fluorescence microscopic pictures (Fig 5) show no fluorescence in these areas, which makes high contents of organic matter unlikely. Results from Wanamaker et al. (2009) suggest that organic matter in *Arctica islandica* is fluorescent under the excitation wavelengths used in our work [55]. Similar results are also reported by Mahé et al. (2010) who found clearly visible growth lines under fluorescent light in *Cerastoderma edule* (460–490nm excitation) [60].

The Mg/Ca ratio of calcite is expected to increase with temperature as the substitution of Mg for Ca is an endothermic reaction ($\Delta H > 0$) [61], the correlation of growth lines with Mg/Ca minima would therefore indicate that they are formed during times of low temperature. There are 20 to 30 microgrowth lines within one yearly increment (Fig 4). This results in a periodicity of 12–18 days, corresponding to a fortnightly growth rhythm. A REDFIT spectral analysis conducted on current velocity data from the Sula Reef revealed significant periodicities of 15d and 4.5-2d (Fig 6). The 15d periodicity fits with the number of microgrowth increments and the lunisolar synodic fortnightly tide cycle. The recorded temperature and salinity data display a periodicity of 15 days in the Leksa Reef. While the temperature and salinity periodicity in the Sula Reef is four days longer than the tidal cycle, these periodicities are generally in acceptance with the growth line periodicity displayed by the bivalve shells (Fig 6). Therefore, it is reasonable to assume that minor growth lines are caused by changes in the current velocity regime of the environment the bivalves live in, possibly coherent with internal neap/spring tide cycles [62]. Internal tides might regulate the food availability [63] or induce growth line

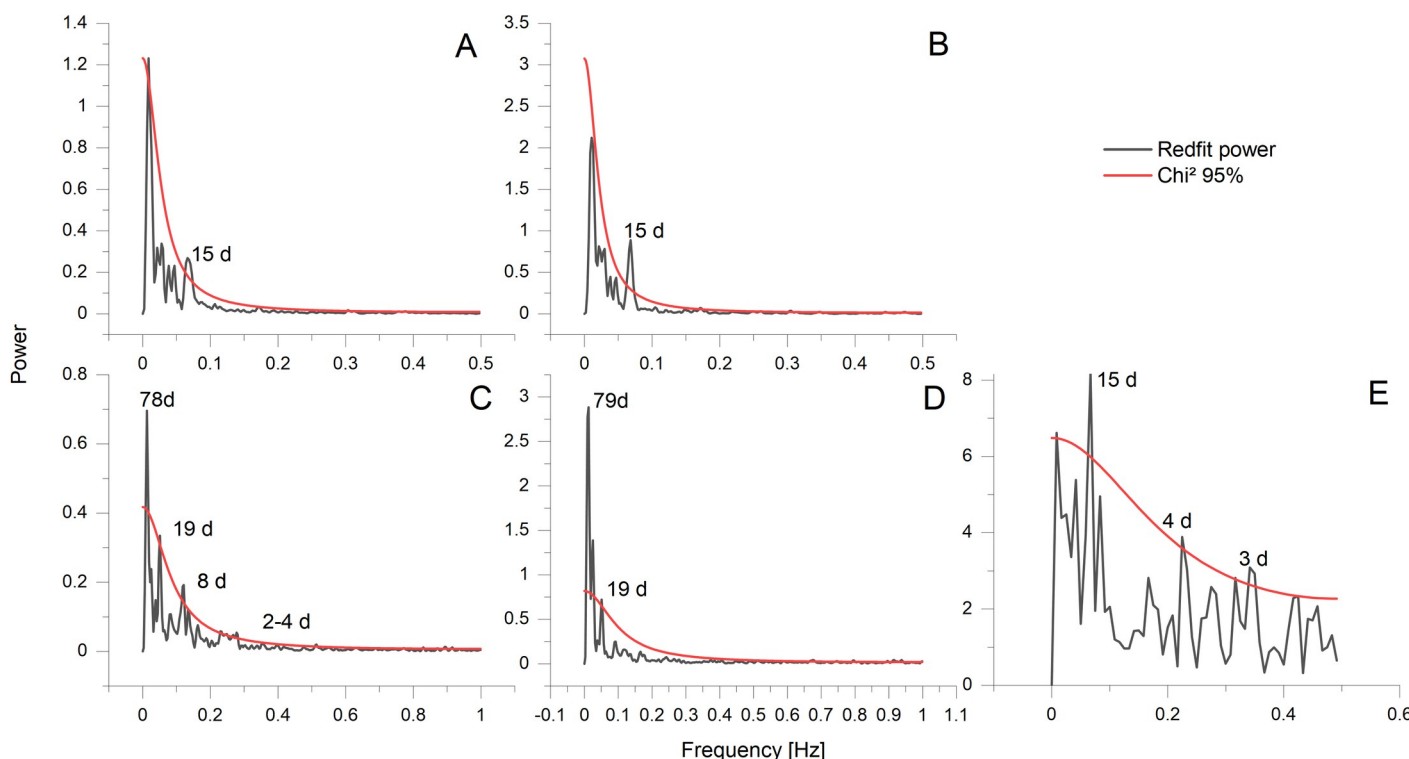

**Fig 6. REDFIT spectral analysis of oceanographic data.** Upper row: Temperature (A) and Salinity (B) at Nord-Leksa Reef. Lower row: Temperature (C), Salinity(D), and flow velocity (E) at Sula Reef.

formation by changing the ambient water mass of the organisms through changes of water temperature [64, 65], salinity [66] or other parameters [67, 68]. Tide controlled growth line formation is also known from several other bivalve species including *Phacosoma japonicum* [69], *Chione cortezi* [70] and *Clinocardium nuttalli* [71]. *Phacosoma japonicum* and *Clinocardium nuttalli* build the growth lines as a response to aerial exposure due to low tides. *Chione cortezi* builds these growth lines due to low tides and consequent increased temperatures, without aerial exposure [69–71].

## Differences between shell layers

Differences between the two observed shell layers can be observed for Mg/Ca and Na/Ca, whereas Sr/Ca ratios show no reoccurring pattern (Fig 4 and Table 3). Differences in the elemental composition between different shell layers are often accounted to differences in the crystal size and form [72, 73]. In *A. excavata* the outermost calcitic layer consists of 50 μm long and 5–10 μm wide prisms. The microgranular crystal in the inner calcitic layer are usually smaller than 1 μm [27]. It is suggested that a significant proportion of Na is absorbed to the crystal surface [74]. This would predict higher Na/Ca ratios in the microgranular layer than in the fibrous layer, as these crystals have a higher surface to volume ratio [73]. Indeed, we observe these higher Na/Ca ratios in the microgranular shell section. This is different for Mg as we observe higher ratios in the fibrous shell layer. Surface adsorption can therefore not explain the higher Mg/Ca ratios in the fibrous shell layer. Alternatively, sector zoning is proposed to explain compositional differences between shell layers consisting of different crystal forms [75]. This effect results in a different incorporation potential for trace elements based on the crystallographic surface [75]. In our case, the sector zoning model would predict a lower variability of Mg/Ca in the microgranular shell layers compared to the fibrous shell layers, due to the more uniform crystals. While we do observe such behavior, the effect is too weak to draw any finite conclusions. On the other hand, it is also possible, that the Mg/Ca ratio dictates the crystal form. It was proposed that Mg poisons the sideward growth of crystals, leading to a more elongated growth [76], which is observable as the fibrous growth in the outer shell layer.

## Effect of $H_2O_2$ on bulk shell material

The purpose of $H_2O_2$ treatment on carbonates is to remove organic material that could potentially alter geochemical measurements [51, 77]. This process is claimed to influence Mg/Ca ratios as the organic matrices in bivalve shells are reported to be rich in magnesium [30]. However, we observe no such effects in the Acesta shells we studied for Mg/Ca. Only Na/Ca ratios are decreasing as an effect of the $H_2O_2$ treatment. While a sodium enrichment in organic rich zones is supported from other organism groups (corals and foraminifera [42, 78, 79]) the limited amount of samples used here does not allow us to draw any strong conclusions. It is also proposed that distilled water leaches sodium which is not structurally bound in the lattice [80]. Based on the differences we observed between the fibrous and microgranular shell layer we expect a substantial amount of surface bound sodium in the fibrous shell layer. The decreasing Na/Ca ratios after $H_2O_2$ treatment can therefore also be caused by the leaching of this surface bound sodium. Since we conducted a cleaning ablation prior to the measurements it is also possible that the areas affected by the oxidative cleaning were already ablated during the cleaning ablation. In this case, however, no changes of E/Ca ratios should be visible.

## Environmental control factors on element/Ca ratios

*A. excavata* shows a larger range in Mg/Ca ratios compared to other bivalves from the subclass of Pteriomorpha, such as *Pinna nobilis* (20.3–29.5 mmol/mol (ICP-AES) [81]) and *Pecten*

*maximus* (5.0–18.4 mmol/mol (ICP-AES) [33]), although we note that these studies did not present highly spatially-resolved analyses such that any such heterogeneity may simply not be present in the data. The bivalve *Neopycnodonte zibrowii*, which lives in a similar setting as *A. excavata* shows similarly high mean Mg/Ca ratios (22.52 ± 17.61 mmol/mol) but higher maximum values of up to 90 mmol/mol, attributable to high concentrations of organic material [82]. The factors that control Mg/Ca ratios in bivalves are not yet entirely resolved. In general, Mg/Ca ratios in bivalves appear to be partly controlled through the calcification temperature but strong (e.g.) kinetic effects inhibit the use of Mg/Ca ratios for environmental reconstructions [32–36]. Similarly, our results show that variations in Mg/Ca of *A. excavata* cannot be explained solely by changes in seawater temperature or salinity. Only 20% of the variability in Mg/Ca can be explained by temperature in the Sula Reef species while it even less (6%) when accounting for all observed specimen from Sula and Leksa. As for many other bivalves, Mg/Ca ratios in *A. excavata* may therefore not be an ideal choice for temperature reconstructions Equally, salinity cannot explain a substantial amount of Mg/Ca variability. On the contrary, multiple linear regression models with temperature and salinity as independent predictor variables together can explain up to 79% of the Mg/Ca variability. This is also true for Na/Ca and Sr/Ca where temperature and salinity can account for 66% of variability, respectively 59%. The different slopes of the regression in the investigated samples, however, show that there is no mechanistic explanation for the correlation.

Kinetic effects are also evident for Mg/Ca and Sr/Ca as shown by the correlation with the linear extension rate. In conclusion these results indicate that Mg/Ca, Na/Ca and Sr/Ca are unlikely to be useful for environmental reconstructions. While environmental parameters such as temperature and salinity certainly have influence on the elemental composition of the shell, strong kinetic and/or biological effects mask these controls.

## Further mechanisms potentially influencing Mg/Ca ratios

**Linear extension rate.** The strongest control on Mg/Ca and Sr/Ca is provided by the linear extension rate which is in accordance with results from other bivalve species [29, 83]. As this effect is not visible in inorganic precipitated calcite [84, 85], it must be caused by the bivalves biological functions [86]. Potentially, the control is not provided by the growth or calcification-rate of the organism, but instead by the metabolic activity and the amount of organic material in the shell [86]. Calcification rate effects on Na/Ca are reported from inorganic precipitation experiments [87], whereas we do not observe such behavior in the bivalve calcite [74]. Again, opposing metabolic effects possibly mask these effects [88]. This will be discussed in greater detail in the following sections.

**Organic material.** High Mg/Ca ratios could possibly be caused by organic matrices, since these matrices may be characterized by high concentrations of magnesium [30]. Because parts of the organic matrices, that act as a framework during carbonate precipitation and mediate mineralization [89, 90], are embedded in the calcium carbonate skeleton [30], geochemical measurements using laser ablation may be affected by organic matter due to different chemical compositions [91]. Organic matter is strongly enriched in magnesium and manganese [30, 92]. Other elements, like barium, are less impacted by the presence of organic material [92]. However, these results are derived from the bivalve *Corbula amurensis*, which shows a much higher organic content than *A. excavata* (19% vs. 1.5%). With 1.5–1.8 wt% (Table 1) *A. excavata* shows organic concentrations, which are at the low end of Bivalvia (*C. virginica* 2.6 wt% [93], *A. islandica* 10.33% [30], *C. amurensis* 19.8 wt% [92]). Given the low organic content in *A. excavata* in the investigated outer shell layer, we expect only minor alterations. While organic matter is usually enriched along major growth lines, our data shows that the high Mg/

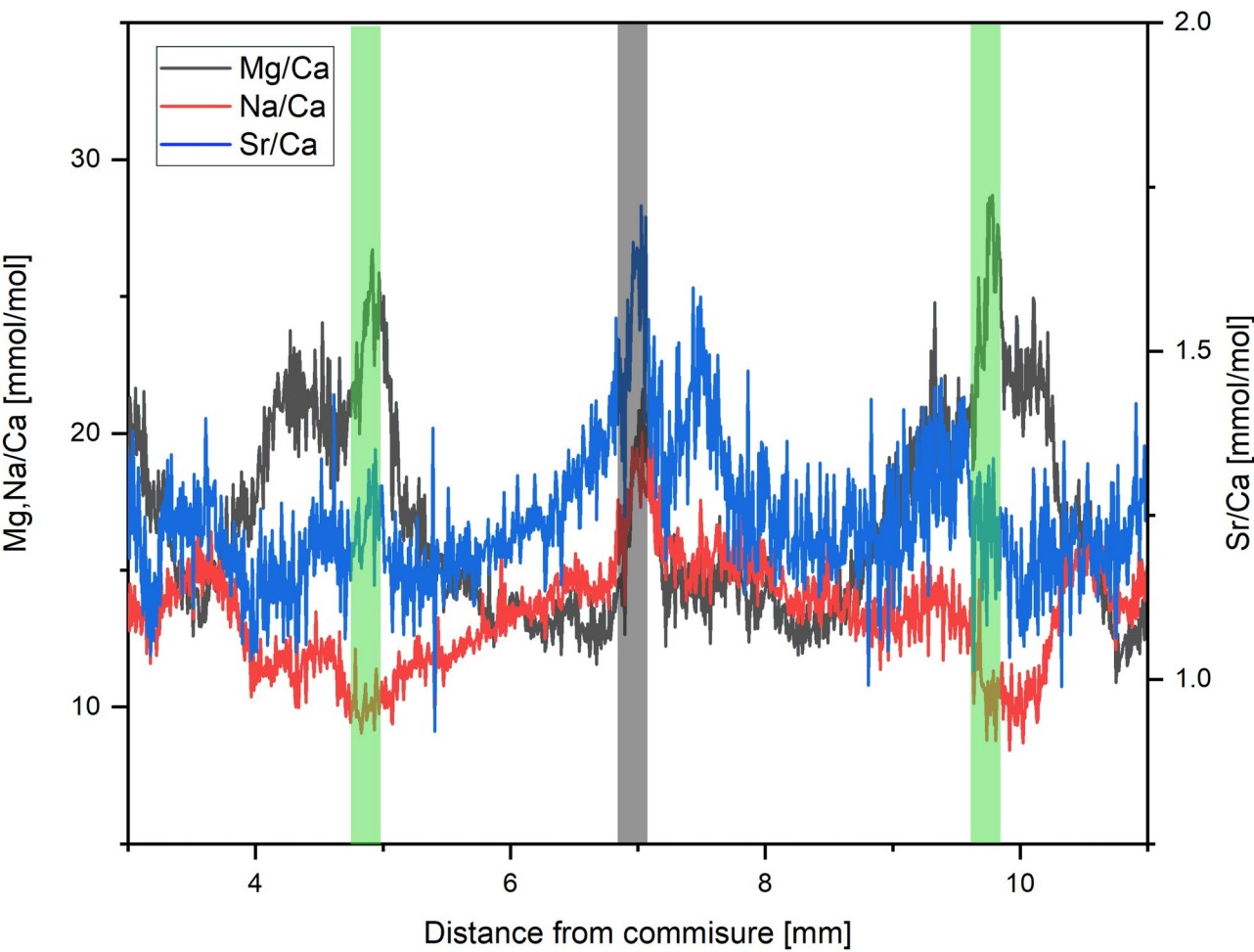

**Fig 7. Mg/Ca Sr/Ca and Na/Ca ratios of sample 6R.** The grey shaded area shows the location of a growth line, which demonstrates increases in all elemental ratios. Green shaded areas show high Mg/Ca winter values. Here increases of Mg/Ca and Sr/Ca are visible whereas Na/Ca is decreasing.

Ca ratios (>35 mmol/mol) of *A. excavata* cannot be accounted for by these organic-rich growth lines. We do note that in the vicinity of major growth lines Mg/Ca ratios irregularly show slight increases by up to 8 mmol/mol, which correlate with increases in Na/Ca and Sr/Ca ratios (Fig 7). Schöne et al. (2010) demonstrated that strontium is enriched in the organic matrix of bivalves as well as magnesium. We are not aware of bivalve specific sodium enrichment in organic rich zones but Na/Ca ratios typically increase in organic rich regions in organisms such as foraminifera and CWC [42, 78]. Protein bound sodium and consequent enrichment of sodium in organic shell matrices is therefore possible, which is also supported by other studies [79, 94]. Even if the amount of sodium bound to organic material might be small (10% in corals [79]) organic compounds such as malate or citrate in the parent solution can increase the amount of sodium coprecipitated in calcite, which would lead to similar effects [95, 96].

**Stress and metabolic activity.** High Mg/Ca variations not accountable to temperature variability are also reported for other bivalve species [35, 74]. *M. edulis* and *A. islandica* showed 14 times and 3 times increased Mg/Ca ratios in temperature-controlled aquarium experiments after handling them for size measurements, which was considered as a stress response by the authors [35, 74]. The underlying mechanisms for this process are yet to be studied, but could

be related to a temporal breakdown of Mg-regulating mechanisms [74] or changes in metabolic activity. In the natural environment stress related responses could be triggered by the influence of changing water masses that introduce large changes in the flow velocity regime. Recorded lander data from the Sula Reef shows flow velocity changes from a yearly mean of 7.8 cm/s up to 150 cm/s in the winter months when also the highest shell Mg/Ca ratios are measured (Fig 3). The increased flow velocities can lead to decreased food concentrations in the water and consequently induce a nutrient deficit in the bivalve [97]. The effect of alterations in the current speed on physiological functions of bivalves was also shown by an increase in heart rate in *Mya arenaria*, which may be linked to an increase in metabolic activity [98]. Mg/Ca and Sr/Ca ratios in the shells as well as in the extra pallial fluid (EPF) increase with higher metabolic activity [86, 99]. We observe strongly increasing current velocities in combination with increasing Mg/Ca ratios and Sr/Ca ratios. Thus, we assume that the observed, increasing Mg/Ca and Sr/Ca ratios in the precipitated shells are the result of a stress-related increase in metabolic activity due to high flow velocities [100], which likely induced an increased influx of Mg and Sr into the EPF. Na/Ca ratios can be altered by changes in the metabolic activity [101] or $Na^+/K^*$ exchange proteins, which are controlled by the metabolic activity could limit the influx of sodium into the EPF and consequently change the shell signature [88]. This is also supported by measurements of the chemical composition of the EPF. During resting periods (low metabolic activity) sodium concentrations in the EPF are higher than during periods of increased growth (high metabolic activity) [102].

Thermal stress may be an additional factor for increasing Mg/Ca and Sr/Ca ratios in the warm winter months. Peak bottom temperatures of 8–9°C in the study area in winter are presumably not problematic for this species. However, short-term variations from their usual adapted temperature may result in stress-induced increases in Mg/Ca and Sr/Ca, but no experimental studies have been conducted to test the thermal tolerance of *A. excavata* so far.

The usual mechanism of bivalves to survive and cope with stress situations is to close their shells, thereby reducing the connection between the living organism and the surrounding medium to a minimum. Shell closure induces a series of consecutive effects on the bivalves such as changes in their heart rate [103], accumulation of $CO_2$ and decrease of $O_2$ concentrations resulting in acidosis of the body fluids [103, 104] and increases in metabolic waste products such as ammonia [104]. Acidosis could have effects on the geochemical signatures in the shell through the buffering process of the body fluids. An increased Ca concentration was measured in both, mantle cavity fluids and EPF in several bivalve species after shell closure [98, 105–107]. Experiments with radioactive $^{45}C$ have shown that the calcium is provided by dissolution of the inner shell surfaces [106]. Dissolution should therefore lead to an increase of all elemental ratios. We do not observe such effects, nor does Wanamaker et al 2019 [35]. A significant contribution of this effect to shell E/Ca ratios can therefore likely be excluded.

## Mineralogical influences on Sr/Ca ratios

Large ions such as strontium and barium are incompatible in calcite, because they cannot easily substitute for calcium due to the differences in ionic size [108, 109]. High concentrations of the small magnesium ion (0.72 Å [110]) that are incorporated can distort the crystal lattice, which increases the size of calcium lattice positions and allows for an increased incorporation of larger ions such as strontium (1.18 Å [61, 110]) and barium (1.35 Å [110, 111]). Based on the relationship between [Mg] and distribution coefficient ($Kd_{Sr}$) given by Mucci and Morse, (1983) from inorganic precipitation experiments, we can calculate the variation of Sr/Ca that can be caused by the lattice deformation that may be induced by changes in shell Mg/Ca alone [61]. The observed variation in [Mg] would result in a predicted $Kd_{Sr}$ increase from 0.159 to

0.228. Accordingly, 0.6 mmol/mol of the Sr/Ca variation could be explicable through Mg-induced lattice distortion, which is in the range of the observed Sr/Ca variation of 1.0 mmol/mol. Individual samples show a good acceptance between [Mg]-predicted Sr/Ca ratios and observed Sr/Ca ratios, which shows that this mechanism can explain a large part (~50%) of the variance in the Sr/Ca data.

## Conclusion

This study represents the first geochemical investigation of the deep-water bivalve *A. excavata*. Results of high-resolution LA-ICP-MS-derived Element/Ca profiles along the fibrous shell section indicate that elemental ratios in *A. excavata* are not well suited for paleo reconstructions due to weak correlations between proxy trace elements and the environmental variables temperature and salinity, which is likely a result of biological vital effects. One of these vital effects might be the here observed correlation between Mg/Ca and linear shell extension rate, which suppresses the correlation with environmental variables.

The growth line periodicity suggests a control of the bivalve growth rhythm through internal tidal waves. These results provide an important indicator to investigate for the distribution of CWC reefs in the past as internal waves are an important distribution mechanism for nutrients in CWC reefs [112].

We propose that the high Mg/Ca ratios in combination with high Sr/Ca ratios and low Na/Ca ratios that occur during winter are caused by combined effects of increasing temperature and salinity together with an increased metabolic activity due to stress. Mg/Ca peaks occurring during summer and in combination with increasing Sr/Ca and Na/a ratios are most likely an effect of a higher concentration of organic matrices. All investigated elemental ratios are known to increase in organic material and the location of these features is in acceptance with the distribution of growth lines, where organic material is concentrated.

The effect of oxidative cleaning with $H_2O_2$ on Na/Ca ratios can be ascribed to a leaching effect on surface bound sodium. The oxidative cleaning did not necessarily remove organic matter as there is no effect on Mg/Ca.

In conclusion it can be stated that Mg/Ca, Sr/Ca and Na/Ca ratios in *A. excavata* are unlikely to be good proxies for temperature and salinity reconstructions at this point, due to kinetic and biological effects on the composition of *A. excavata* calcite. Fully controlled cultivation studies are needed to gain a thorough understanding of the factors that influence element incorporation. In Combination with additional techniques such as clumped isotopes, vital effects might be accountable for and Acesta excavata might offer a high resolution archive for the reconstruction in deep-water coral reefs.

## Supporting information

**S1 Appendix. LA-ICP-MS results (all shell layers).**
(XLSX)

**S2 Appendix. LA-ICP-MS results (fibrous shell layer).**
(XLSX)

**S3 Appendix. LA-ICP-MS results (microgranular shell layer).**
(XLSX)

**S4 Appendix. LA-ICP-MS results ($H_2O_2$ treatment).**
(XLSX)

**S5 Appendix. Correlation of elemental ratios to environmental data.**
(PDF)

**S6 Appendix. Environmental data.**
(XLSX)

**S7 Appendix. Shell pictures and geochemical data.**
(PDF)

**S8 Appendix. Intra-reef variability of E/Ca ratios.**
(PDF)

## Acknowledgments

We are grateful to all cruise captains, crew members and cruise participants of research cruises POS455, POS473. We are also grateful for the comments of seven anonymous reviewers. This is FIERCE contribution No. 54

## Author Contributions

**Conceptualization:** Jacek Raddatz, Max Wisshak.

**Data curation:** Nicolai Schleinkofer.

**Formal analysis:** Nicolai Schleinkofer.

**Funding acquisition:** Jacek Raddatz.

**Investigation:** Nicolai Schleinkofer, Jacek Raddatz, David Evans, Axel Gerdes.

**Project administration:** Jacek Raddatz.

**Resources:** Jacek Raddatz, Axel Gerdes, Sascha Flögel, Silke Voigt, Janina Vanessa Büscher, Max Wisshak.

**Supervision:** Jacek Raddatz.

**Visualization:** Nicolai Schleinkofer.

**Writing – original draft:** Nicolai Schleinkofer.

**Writing – review & editing:** Jacek Raddatz, David Evans, Axel Gerdes, Sascha Flögel, Silke Voigt, Janina Vanessa Büscher, Max Wisshak.

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
