## [Decision Letter · Decision Letter 0]

30 Jun 2020

PONE-D-20-12353

Compositional variability of Mg/Ca, Sr/Ca, and Na/Ca ratios in the deep-sea bivalve *Acesta excavata*

PLOS ONE

Dear Mr. Schleinkofer,

Thank you for submitting your manuscript to PLOS ONE. The reviews that I have received indicate that the manuscript has merit, but does not meet PLOS ONE’s publication criteria as it currently stands. Therefore, we invite you to submit a revised version of the manuscript that addresses the points raised during the review process.

I have received a total of 5 reviews. One was a preliminary read of the paper that recommended rejection because of limited data, but also suggested that I ask for recommendations from reviewers with more experience with deep sea bivalves. A second reviewer had seen the prior version of the manuscript and recommended minor revisions based upon improvements that you had made since review of the prior version. Three other reviews that I solicited all recommended major revisions be made prior to acceptance. I think from reading the reviews that you have received a lot of good suggestions with both specific recommendations, as well as more general critiques on how to better organize the paper and present your data. All of these substantive reviews are constructive and I would ask that you carefully see if you are able to revise the paper to address these recommendations. Please also note the requests to make the data available upon publication and other journal requirements.

We look forward to receiving your revised manuscript.

Kind regards,

Lee W Cooper, Ph.D.

Academic Editor

PLOS ONE

"JR and NS acknowledge funding from DFG project RISE RA 2156-5. FIERCE is financially supported by the Wilhelm and Else Heraeus Foundation, which is gratefully acknowledged. This is FIERCE contribution No. 14"

4. We note you have included a table to which you do not refer in the text of your manuscript. Please ensure that you refer to Table 1 in your text; if accepted, production will need this reference to link the reader to the Table.

Reviewers' comments:

Reviewer's Responses to Questions

**Comments to the Author**

1. Is the manuscript technically sound, and do the data support the conclusions?

Reviewer #1: Yes

Reviewer #2: Partly

Reviewer #3: Yes

Reviewer #4: Yes

Reviewer #5: Partly

2. Has the statistical analysis been performed appropriately and rigorously? 

Reviewer #1: I Don't Know

Reviewer #2: No

Reviewer #3: Yes

Reviewer #4: Yes

Reviewer #5: No

3. Have the authors made all data underlying the findings in their manuscript fully available?

Reviewer #1: No

Reviewer #2: No

Reviewer #3: Yes

Reviewer #4: Yes

Reviewer #5: No

4. Is the manuscript presented in an intelligible fashion and written in standard English?

Reviewer #1: No

Reviewer #2: No

Reviewer #3: Yes

Reviewer #4: No

Reviewer #5: No

5. Review Comments to the Author

Reviewer #1: I am uncertain that the small number of specimens is enough to draw any conclusions here.

Also the small variability in temperature seems far too minute to get any good patterning especially if the vital effects are not clearly understood.

But I abstain from any judgement as I am not familiar with deep sea bivalve records.

Reviewer #2: This is an interesting study with some intriguing results that the paleoceanography and sclerochronology communities would find useful. That said, the paper lacks structure and direction and the reader does not know what the study did until they read the results, and even then it is still confusing. Jumping to correlations and scatterplots is leaving a lot of great and interesting parts of this study as side notes. The problem starts with an introduction that doesn’t relate to the study and its findings. This is an archive-proxy development paper. Tell the reader what and why you are testing to establish this archive-proxy in the introduction. For example, testing the cleaning methods to see if they change the shell geochemistry, excellent but its meaning is lost in the paper. Testing the difference between the fibrous (black line) and microgranular (red line) shell sections, excellent but its meaning is lost in the paper. Assessing multiple shells are two different sites for reproducibility, excellent but its meaning is lost in the paper. And so on. Then structure the results, discussion, figures and tables, to report these findings. See Wanamaker 2019 https://doi.org/10.1016/j.chemgeo.2018.02.012. May I suggest looking at some other coral replication and archive-proxy papers for guidance on how to structure and report the results from this study. See Stephens 2004 doi:10.1029/2004GL020343, Smith 2006 www.agu.org/journals/pa/pa0601/2005PA001187/, Wu 2014 doi:10.1016/j.palaeo.2014.07.039, DeLong 2016 doi:10.1016/j.palaeo.2016.08.028. Also see review of Jones 2009 doi:10.1177/0959683608098952.

The beginning of the introduction does not appear to be relevant based on the abstract, some very wide and irrelevant statements are made. It gets better after line 37 and reference 5. For the introduction, do not just make broad statements that are generally true but may have no relationship to your conclusions and your study. Write your introduction so that it provides the reader with the background information and motivation for your study so that the introduction supports your findings and conclusions. It is OK to say “there are not many immediate water archive-proxies and you are exploring the feasibility of this new archive and its elemental proxies”. I suggest removing the sentences from Line 30-37. Start with cold water corals being sensitive and expand from there.

At the end of the introduction lines 87-92, the authors state what this study will show. This study has in situ environmental data for one year for comparison to their shells! This is impressive and is needed for developing a new archive-proxies. This is the point you should emphasize in your introduction. Talk about how hard it is to get instrumental data where your archive lives, why it is important, and this helps your proxy development. Revise your introduction to support all the different parts of your study, why those assessments are important and different from previous studies. See the review of Jones 2009 that summarizes the status of High-resolution palaeoclimatology as a guide for your paper. You will notice mollusks are not even mentioned, and this review paper is only 11 years old. Look at the coral archive-proxy development and what is needed and you can use that paper a blueprint for what needs to be done to develop other annually-resolved marine archives and proxies.

This study has in situ temperature and salinity. This is key part of the study and makes this study unique. However, there are some key details left out. How often were the measurements taken? hourly, daily? The authors also mention ARGO data was used, but this data and the in situ data are not shown in the paper. Furthermore, the authors never discuss how time in the shells was assigned for comparison to the in situ instrumental data. This is important, and will change all the correlation results, among shells and a with shells and instrumental data.

Methods: How was soft body removed from the dry shells? Sonication, ultrasonic cell disruptors? Were the slabs in epoxy cut more? If so how? What precautions where taken to ensure cleaning and polishing were not altering shell chemistry? Can you give the water depth the shells are from and depth of the landers recording ocean conditions?

The standards for LA-ICP-MS accuracy and precision: The accuracy is a large number, was the data adjust for the accuracy offsets? What Ca isotope was used for Elemental/Ca? Was the same Ca isotope also used for the internal standard? How was the ICP-MS data processed? Many labs use WaveMetrics IgorPro with Iolite software.

The study only samples a small part of the shell, can the whole shell be sampled and how old does this species get?

Detailed items: Note some of these items (spelling, grammar, useage, etc.) are reoccurring problems in this paper. The authors should use find and replace to fix throughout the manuscript and carefully edit and revise.

Line 11 revise to “…cold-water coral reefs…”

Line 13 revise to “…instrumental seawater parameters such as…”

Line 15-17. This is redundant, if correlation is weak, the explained variability will be low. R^2 = variability explained. Revise to be more concise and give the r value, with the n and p-values. Do this for all reported correlations throughout the paper (such as lines 18, 19). Same for sensitivity, put this value in parenthesis with its uncertainty, so the reader knows what you are referring to.

Line 20 Do you mean the correlation of elemental ratios with extension rate was the highest? The “control” is assumed, correlation is not causation. Revise.

Line 21 The term “vital effects” is used by many authors and the definition is not clear, it could mean biological effects, reproduction effects, species effect, depth effects, calcification rates, etc. Be more specific, just say “growth-related effect”, determine by linear extension. Line 23-25 shows that these “vital effects” are quite varied.

Line 22 Clarify “Multiple linear regressions” did you do regression many times or once? If once revise to “Multiple linear regression”.

Line 22 “a large amount of elemental variability” Be specific and give and exact number.

Additionally throughout the paper, avoid using descriptive adjectives (i.e. large, low, etc.) for numerical analysis. These adjectives have no meaning, what value do you assign to “large” or “low” better to give the value and let the reader decide if the value deserve the descriptor. This removes any bias the authors place on their data.

Line 23 Calcification rate does not equal linear extension rate. Calcification rate = linear extension rate x density. How do the authors account for density differences that would impact calcification rates?

Line 26-28 This needs to be explained better. Just using Mg/Sr does not mean you get around growth-related effects. Revise “combining different proxies such as Mg/Sr ratios” Proxies are some property of an archive that can be a “proxy” for some environmental parameter of interest. You mean “combining different elemental concentrations such as the ratio of Mg to Sr…” Stating Mg/Sr ratio is like saying ratio twice since it is given as a ratio with /.

Line 30 Revise “Intermediate water masses have been shown to respond with highly sensitivity to climate forcing”

Line 32 use a comma before “which”, always. Fix throughout. Additionally, double check the “that” and “which” are used properly each time.

Line 31-33. This sentence confuses me. Be more specific. What do you mean by “atmospheric climate changes” and “small scale effects” ? CO2? Water vapor, wind? Waves?

If your archive-proxy is intermediate water why not just say they reflect intermediate water oceanic conditions. From your abstract, the proxies are argued to be co-varying with water temperature and salinity. What does these two properties have to do with the atmosphere?

Line 47-48 This sentence is confusing, revise “Water flow velocity and indirectly also its effect on food supply to the corals provide further controls.” Something is missing after “indirectly”. What does “its” refer to? Perhaps the “also” is not needed, makes more sense without also? What is being controlled? In general, avoid using “it” and its forms, be precise and tell the reader what you are referring to. Remove “it” throughout by revising the sentences.

Line 52-53 Shallow water reef corals spawn with Moon phases. I would not be surprised if cold water corals do the same, see link.springer.com/article/10.1007/BF00301193.

Line 56 The reference refers to just one cold water coral species Lophelia pertusa, and there are jundredss of these corals, so how can you extrapolate from one species to all of them? Less work has been done with cold water corals but does not mean they all do not make good archives of past conditions. This sentence is also missing a period at the end. Most of your references are for L.pertusa but there are other cold water corals that can be used for reconstructions such as bamboo corals (see doi:10.1016/j.gca.2018.12.027, doi:10.1016/j.epsl.2019.11575, doi:10.1029/2011PA002260, doi:10.1029/2010GC003443, www.sciencedirect.com/science/article/pii/S0016703711002808, Desmophyllum www.sciencedirect.com/science/article/pii/S0012821X12003652, Adkins 1998 www.sciencemag.org/cgi/content/abstract/280/5364/725, dx.doi.org/10.1016/j.gca.2011.02.019, www.sciencedirect.com/science/article/B6V66-50PVG4J-2/2/4a35c81fe923000631c5a6f4b211ef50 and others.

Line 57-58 Reference #25 is not a coral paper but an interlab comparison paper and not an appropriate reference.

Line 58-59 I do not argue that bivalves are not valuable archives for reconstructions. Yet deep sea corals are still a work in progress and it is not time to give up on them. I would also argue they have simple growth patterns like a tree-rings. Additionally, bivalves tend to grow slower in colder water making seasonal reconstructions more difficult except for the intervals in the youngest parts of the shell where growth is the fastest or using laser ablation ICP methods, which can also be used with deep sea corals. As the shell ages, the seasonal resolution becomes more difficult to obtain and many researchers use annual resolution to produce long continuous reconstructions or extension rate themselves for the reconstruction. Be careful of negatives you describe for any archive-proxy because they can also impact your study!

Line 61-65 What about bivalves from the tropics and subtropics? They still have growth lines and they can be annual. Give a reference that states growth lines are annual. Additionally, if you just refer to the species in your study, that would be better than generalizing for the tens of thousands of bivalves species that exist. Do all bivalves have visible growth lines and all they all annual? I bet I could find some that are not. Revise to be more specific and do not make class wide generalizations.

Line 65-66 This is an incomplete and confusing sentence “First observation make a seasonal accumulation of growth increments in A. excavata likely.”

Lines 66-69 Another sentence needing revision for clarity, suggestion “Different stages of growth increment development in A. excavate, which were collected at different times throughout the year, have been observed, as well as cyclic density changes in the shell, which are possibly controlled by temperature or salinity changes.

Line 70-71 Reference needed. doi.org/10.1016/j.chemgeo.2019.01.008 suggested otherwise.

Line 78-81, For Na/Ca also see www.sciencedirect.com/science/article/B6V6R-4894W8C-89/2/9bfe9e583e5fa7fc01c95e81a6604064, and doi:10.2343/geochemj.1.0067.

Line 86 delete “very”. This word should be avoided in technical writing unless it adds to the sentence meaning. Very is an overused word. Whenever you’re tempted to use it, try dropping it to see if any meaning is lost.

Line 105 Do you mean the slab surface was polished? How were the surfaces polished?

Line 109 and figure 1. Use of “commissure”, do you mean where the two valves meet or the shell edge? This term is usually used with brachiopods. Just say ventral margin or edge of the shell.

Line 100 What is “the ear of the bivalve”? Bivalves do not have ears. See Bailey, 2009: Journal of Paleontology 83, 493-495.

Line 112 “location marked in A with a dotted square” The top image is marked with an A, but there is no dotted square in in the figure. There is a dotted rectangle inside the solid rectangle, is that what you mean? The image in B looks like the orientation is from the Umbo to the ventral edge not along the growth lines. Use a A-A’ to mark the orientation and location of B.

Line 113 space between number and unit, check throughout. Line 162 space between word and reference, check throughout.

Line 114 Revise “the opposing side”, you mean just the other side, not both sides, correct? Replace “surfaces” with “slab” since that is the term you used earlier or “slab surface”. Does this mean you cut two slabs from each shell or did you just cut them in half? I assume you have to cut thinner slabs to fit in the laser chamber.

Line 114 Define LA-ICP-MS at first use, it is defined in line 147. Same for all abbreviations, check throughout the paper. Line 221.

Line 180 Spelling mistake “recorded”. Cite the figure you are referring to.

Line 181 Summer and winter have more than one month, use “months”.

Line 180-181 Revise “Temperature varies from 7.7°C in the summer months to 8.5°C in the winter months.” The summer is colder than the winter? Odd. Is the Sula reef the same? OK, I see the explanation in lines 184-186, move this up to the temperature sentences.

Line 182 I do not think you mean “The Sula reef shows a temperature variability of 7°C – 8.8°C” meaning the temperature can vary by 7 to 8.8 degrees in a year but the temperature changes from 7ºC to 8.8 ºC during the year. Where are these results and data in the paper? There should be a figure and a data table for these data. If the data is already publicly available, then a citation is needed with the internet access address.

Line 183 Be consistent with to or –. “33.5 to 36 g/kg in the Sula reef and 34 – 36 g/kg in...”

Line 191 revise “The different shell layers are not easily identifiable…” layer is more descriptive of what you are describing.

Line 220-223 Means are not robust against outliers. What do are the median values? Isn’t Fig 2 the data from the shells, not fig 3 which are scatter plots. Is the mean in line 221 for all shells or just one, clarify.

Figure 2 I do not see the yellow line for the laser path. Perhaps use another color like blue?

Figure 3 This is a scatterplot of the elemental data for just two shells. Where is the data for the eight shells, like Figure 2? This needs to be included somewhere in the paper. There are eight shells measured, where are those plots? Establishing the annual cycle in the archive is important, as you point out in our introduction. So where is the proof of annual layers for these shells? At least have a table with the descriptive statistics for each shell, then you do not have the write it out in the results. I cannot see any “hollow” marks mentioned in the caption. How were outliers identified i.e., what statistical test did you use? How were the “fit functions” determined? With this many data points, any correlation will be significant, but not if you assess correlation with degrees of freedom. See Rodgers 1988 www.jstor.org/stable/2685263 and EBISUZAKI 1997 doi:10.1175/1520-0442(1997)010<2147:amtets>2.0.co;2.

Line 226 “A correlation between Mg/Ca ratios and Na/Ca ratios is visible” where is this visible? Visible is not very scientific, give a correlation value with n and p-values. This sentences jumps to Na/Ca which has not been described yet. Talk about correlation among El/Ca after presenting results for each El/Ca.

Line 229 is this for all shells? See previous comments for Mg/Ca.

Line 230 How was no difference determined? What statistical test did you use?

Line 231-233 What statistical test were used to evaluate correlation? Add a correlation table for the shells and El/Ca. For an example of how this can be done see Stephens 2004 doi:10.1029/2004GL020343.

Line 238-239 This is an interesting result, I like that you tested this. How many shells did you test this for? But where is the data and figure? Whenever you use the term “significant” there needs to be a statistical test to determine the significance. How was significance tested? Provide those details.

Line 243 Microgranular shell section Move this section up to the beginning of the results since you already started describing it. How are the growth rates different among these shells? Could growth rates be driving the differences you noted in this section of the manuscript?

Line 252 Correlating element/Ca data to instrumental data

Before jumping to correlation, where are the growth rates of the shells described? I would put all the physical properties of the shells first in the results section. Then geochemical results, cleaning results, replication results – intra and inter reef. Then you can move on to correlation between the geochemistry and environmental data.

One of the largest issue of correlating shell geochemistry to environmental data is time assignment in the archive, this is also a large source of error. Some researchers with correlate or “calibrate” the geochemistry with environmental data using just the max and min of a seasonal cycle -this is the easiest to do. Coral paleoclimatologist use Analyseries or QAnalyseries (sites.google.com/site/geokotov/software) for time assignment of many years to an SST record. Some sclerochronologists use von Bertalanffy growth curve to help with time assignments. Problem with LA-ICP-MS data is you have a lot of data that makes time assignment cumbersome but doable. Please give details on how you are assigning time to your shell data. If you assume Mg/Ca is correlating with temperature and you sue that for your time assignment, it will have the highest correlation! Please state the basis for make this assumption and test doing time assignment with the other geochemistry to see if the results vary (they should). By using Mg/Ca for time assignment, you have bias the correlation analysis. If you can use the growth lines to assign time then you can remove or reduce this bias.

Lines 257-265 Please add a correlation table to show the reader these results. The shells with low correlation, could that be a time assignment issue? Look at replication. Does the correlation improve if you build a master record with two to eight shells? The signal to noise ratio should improve as you average shells together and thus the correlations improve, if there is really a causal relationship.

Line 271-275 now it is mentioned that data were tuned. Shouldn’t this be mentioned before the correlation analysis?

Line 276-282 Can you provide a citation for “nonlinearity of bivalve growth”. Does this happen within a year? Or do you mean on longer time scales? How did you “tune” the data? What software program did you use? Did you interpolate to get the geochemistry to a time scale? Include a plot of your raw shell data and your tuned shell data in your paper or appendix.

Line 280-283 Distance in time-geochemistry or growth lines in the shell? How many years do you have per shell? Maybe 6 from fig 2? Is this enough years to assess a relationship between extension rate and geochemistry statistically? Have you extension data per shell in your paper as well.

Table 2 How was p values assessed? # of data points or degrees of freedom? With LA-ICP-MS data there are thousands of data points so any correlation will be “significant” but you should assess degrees of freedom and re-determine the p-values, see EBISUZAKI 1997.

References need to be editing for spelling, proper capitalization for proper nouns, species names should be italics, [internet] removed and doi given instead. Access dates removed, etc.

Reviewer #3: My edits and suggestions are noted below. I think this version of the manuscript is much improved by not pushing the data too far. The results are interesting and will be useful.

I think that the Introduction section, especially the literature regarding bivalves is out-dated and not very informative. I made a suggestion below on how this might be improved.

Abstract - - “Our results may still hold promise that A. excavata may serve as a paleoceanographic archive

for intermediate water masses despite the relatively strong vital effects, as these can potentially

be accounted for by combining different proxies such as Mg/Sr ratios.” Add a new sentence in the abstract, …more work is needed to confirm this idea.

Line 56- missing period at end of sentence.

Line 58-59- some would argue that a compressing record with ontogeny is not simple.

Line 73- ref 32 is certainly out-dated. The study has never been replicated, either. If you are going to list one study that seemingly work for M. edulis, be sure to include other studies (e.g., Wanamaker et al., 2008) that show that the errors are very large using Mg/Ca thermometry and that vital effects seem to be dominant.

Wanamaker, A.D., Kreutz, K.J., Wilson, T., Borns, H.W., Introne, D.S. and Feindel, S. (2008) Experimentally determined Mg/Ca and Sr/Ca ratios in juvenile bivalve calcite for Mytilus edulis: implications for paleotemperature reconstructions. Geo-Mar Lett 28, 359-368.

Line 78- see discussion by Wanamaker and Gillikin (4.3 The use of element to calcium ratios in bivalves as environmental proxies) on this very same issue. It would be good to cite this study here. And many of the references used there are more up to date. The literature review on bivalves in this section need to be a bit more up-to-date.

Wanamaker, A.D. and Gillikin, D.P. (2019) Strontium, magnesium, and barium incorporation in aragonitic shells of juvenile Arctica islandica: Insights from temperature controlled experiments. Chemical Geology 526, 117-129.

Line 260. Capitalize Line.

Line 399- Should be Wanamaker and Gillikin

Reviewer #4: First, I grateful the chance to read this paper. In this study, Mg/Ca, Sr/Ca and Na/Ca ratios were measured on live-collected bivalve Acesta Excavata (Fabricius, 1779). Even though, none (or very poor) correlation has been found between trace-elements ratios obtained from shells sequentially measured and seawater temperatures and salinity, authors concluded that these proxies have very important implication for further palaeoclimate investigations. In the last section of the discussion, they assume that the vital effect could be removed combining different trace-elements, like for example Mg/Sr, in order to try to extract a clear climate signal from the trace-element series. However, the correlation obtained from Mg/Sr with seawater temperature is still very low (Fig. 7; r2= 0.15). From my point of view, this is the most weakness of the manuscript. This study contains valuable information but the conclusions and the implications for further climate investigations are very few, especially for a journal with a very high prestigious like PlosOne. For that reason, my recommendation is major revision. The should try to explain better what are the application of their results for future studies.

All in all, more minor/major issues should be also considered.

General comments:

- The manuscript is not very well written, and the author should be check grammatical mistakes or inconsistencies. For example: L. 56. Point after reconstruction (it is the end of the sentence). L. 260: Four with capital letter (it is the beginning of the sentence). Same line: You said “four of the investigated samples”, but you only reference three “(12R, 16R, 17R)” and then you included four values. L. 308 and 311. First you wrote “Wanamaker et al.,” and then “Mahe et al,”. Write a point in the second one and remove the comma in both cases. This is not necessary in the text (only if the reference is between parenthesis, which is not the case in this journal). Fix it throughout the text.

- As you can see, my English is not good enough to do any corrections (I apologize if you see any grammar mistakes), but the English of the manuscript does not sound very well. It would be great if a native English speaker could read the manuscript carefully and give you a feedback.

Figures:

- I suggest to add a new figure including a map and a picture in order to show where the shells were collected.

- Figure 2: You must include the series obtained from all shells, not only the results that you have obtained from one of them.

- Figure 3: I guess that they have included dataset from all shells. Is that correct? If not, results obtained from all shells should be added.

- Figure 4: Improve this figure. The resolution is very low. Besides, add the scale.

Specific comments:

- L. 2: Please, add "(Fabricius, 1779)". Authors should include this information in the tittle, in the abstract, as well as the first time that they write this species in the text. They also must include the complete name of the species (Acesta excavata) the first time that they cited this taxon in the abstract and in the introduction.

- L. 13: Please, specify in the abstract what technique you used to measure this trace elements, i.e., LA-ICP-MS.

- L. 62-66: Authors must include more information about growth patterns of this species if this is available. How many days throughout the year this species grows? What is the longevity of this species? Information about the effect of the ontogeny in the growth patterns is available? Indicate when the sexual cycle occurs? Gametogenesis? Spawning? Is there a relationship between sexual activity and growth stoppages?

- L. 90: Please, specify here what technique you used to measure these trace elements.

- L. 105: Please, explain with more details what material and methods you used for polishing shell section. Glass plates? SiC grit powder?

- L. 146: How do you know this calcium carbonate layer is calcite and not aragonite? You must investigate the shell mineral composition, for example using X-ray diffraction (XRD) or, at least, referencing a previous study where calcium carbonate composition of this species would have previously been investigated.

- L. 181: Why do you have warmer temperatures in winter? This is a little strange for me and maybe for other readers too. Please, try to explain this. I think that it is very normal and of course that there is a very simple explanation, but it is strange.

- L. 204: The same that previously. How do you know that layers are composed by aragonite or calcite? Please explain that.

- L. 205-206: “These lines coincide with minima in Mg/Ca ratios, although these lines are not always visible”. The sentence is very tricky for a good understanding. How do you know that minima Mg/Ca ratios systematically match with these lines if they are not always visible? Please, you should explain this better. Why should I suppose then that it is true if you cannot see all lines? That is crucial.

- L. 226: I guess that these results about correlations between trace-element were extracted from figure 3. So, they should reference it in the text.

Reviewer #5: This manuscript’s dataset has potential to make an interesting contribution to investigations of mollusc shell elemental ratios as proxies for reconstruction of seawater parameters (i.e. temperature and salinity).

The key strength of the study is the ability to make comparisons between mollusc shell chemical proxy records and instrumental records of seawater temperature and salinity, as well as current velocity. However, these important oceanographic data are not presented within the manuscript and should be included as separate panels on a new figure. Figure 5 also should be expanded to include time-series analysis of the instrumental seawater temperature and salinity datasets (to complement the current velocity assessment), even if no clear periodicity is event. Without sight of the oceanographic data it is not possible to discern if the shell Mg/Ca records are even tracking seawater temperature. The manuscript also should be explicit with regards to the distance between the lander oceanographic instrumentation and the sampled mollusc specimens, and it is rather unclear why ARGO data are preferred to the available lander instrumentation. A site map, to support text lines 95 to 98, also would be useful.

The key weakness with the manuscript is the approach taken to convert distance along shell growth axes to time, i.e. the clearly demonstrated very weak relationship between shell Mg/Ca and seawater temperature really does not support use of Mg/Ca minima as tie points for age model development, and how interpolation between such points has been undertaken is rather unclear (i.e. constant or variable growth rates). Furthermore, it really does seem to be quite a circular argument to develop an age model using one elemental ratio (i.e. Mg/Ca) and then test for a relationship between that same proxy and seawater temperature. Also, looking at Figure 2 the ‘peak’ in Mg/Ca seems to occur in different positions relative to the ‘assumed yearly growth lines’, seemingly further negating the age model approach that has been adopted. (The yellow dashed line on Figure 2 also is not clear.) Finally, the cross section of the shell shown in Figure 4 apparently shows a clear discontinuity in shell growth, presumably due to a significant external forcing rather than a response to a slower change to external seawater conditions; could such an abrupt forcing have been changing current speed (data are not plotted to assess) or even predation? Cross sections of the other investigated mollusc specimens, even if only included within a supplementary information section, would be useful for comparative purposes and enable a more complete assessment of shell growth variability between the studied specimens.

The text on lines 316 to 333 implies that an age model can be developed using the internal shell increment growth series (as shown in the bottom right panel of Figure 2), such an age model then being entirely independent of the shell elemental ratio proxies and thereby facilitating considerably more robust assessment of any relationships between shell chemistry and seawater parameters. This section, which should be called sclerochronology (and not scleromorphology) should appear after presentation of the oceanographic data and before any discussion of the mollusc shell elemental contents. It should also be possible to include graphs that present a comparison of the shell growth increment sizes (and thus growth) between the individual specimens investigated, to discuss further the comparability between the different shells for the same time period, following presentation of a robust argument for the factor(s) controlling shell increments, e.g. tidal periodicity as is mentioned within the manuscript’s text. Of course, an alternative approach to support development of an independent age model would be via use of shell oxygen-isotope ratios which could then be compared to predicted values derived from the measured seawater temperature and salinity datasets (not many studies are in a position to adopt such an approach, which this dataset actually does facilitate); such an approach also would help resolve the question as to whether there is any seasonal cessation of shell growth in the studied locations (theoretically also identifiable by counting of growth increments between growth lines, if the latter are not formed by irregular events such as predation). Mark and recapture of mollusc shells also would have helped with robust development of age models, but such an activity presumably was not completed in this study?

The value of Figure 3 is very unclear, because the primary interest is whether the shell elemental ratios relate to oceanographic parameters, rather than to each other. Furthermore, other studies have shown that different mollusc species can exhibit differences in elemental ratios between individuals (cultured in the same experimental aquaria), such that merging all specimens together potentially hides inter-specimen variability. In fact, the caption to Figure 3 confirms this to be the case for this dataset, because two specimens are excluded as ’outliers’! If something like Figure 3 is to be presented then there should also be separate panels for each individual specimen. What would be much more useful would be to replicate Figure 2 for each of the other seven specimens, ideally plotted against an independent growth-increment (or oxygen-isotope ratio) derived age model to test whether multiple specimens from the same location(s) faithfully record consistent elemental ratios through time. Only one specimen is shown in detail (in Figure 2) at present, and plotting of elemental ratios for all individuals against time also would facilitate testing of whether the shell chemistry better follows ambient oceanographic forcing at certain times of the year, which cannot be discerned when the data are lumped together as is the case in the current manuscript. Note: a supplementary information file can be used to include multiple figures, if they will not fit within the main article.

Overall, there is likely considerably more variability and interest within the dataset than is discussed at present. For example, once independent age models are developed, Figure 7 should also plot the relationships for individual specimens and not just all specimens merged together for the two reefs. Statistical testing also could be used to test for differences between specimens, and between the two sampled reefs. Furthermore, from Table 1 it is evident that all eight specimens investigated were of a comparable size (ca. 100 mm in length) and thus all adults. Could data also be included for some smaller, i.e. juvenile, individuals, so that an assessment could be made as to whether ontogeny has any influence on elemental ratios within this species? Another more minor point, was any shell periostracum removed prior to determination of shell organic content? And, can parts of Figure 4 be presented at higher magnification?

Finally, the current Discussion section is not that coherent; there are long paragraphs that contain multiple short statements that are not very well interlinked. Some statements also seem rather too speculative, such as in relation to selective dissolution of calcium from a mollusc shell? Ultimately, the relationships between elemental ratios and oceanographic parameters (i.e. temperature and salinity) are very weak and thus of limited utility. The manuscript tends to overstate what might be possible, e.g. arguing that Mg/Sr ratios are better than Mg/Ca, but the r-squared values (for all data combined) only improve from 0.07 to 0.15 (from Figure 7)! There could well be more significant relationships embedded within this dataset, but such is not evident from the approach that was taken to merge all data values together.

6. PLOS authors have the option to publish the peer review history of their article (what does this mean?). If published, this will include your full peer review and any attached files.

Reviewer #1: No

Reviewer #2: No

Reviewer #3: No

Reviewer #4: No

Reviewer #5: No

---

## [Author Response · Author response to Decision Letter 0]

5 Oct 2020

We are thankful for the comprehensive reviews we received. They were very helpful and helped us tremendously in improving our manuscript. We reworked large parts of the manuscript and incorporated the reviewers remarks as best as we could. 

Reply to Reviewer 1

Rev: I am uncertain that the small number of specimens is enough to draw any conclusions here.

Also the small variability in temperature seems far too minute to get any good patterning especially if the vital effects are not clearly understood.

But I abstain from any judgement as I am not familiar with deep sea bivalve records.

Author comment: We thank Reviewer 1 for his review. With 4 specimens per location, we have a similar amount of specimens as other comparable studies ((Lazareth et al., 2013; Marali et al., 2017a; Schöne et al., 2011)

Reply to Reviewer 2

Author comment: We thank the reviewer for these very helpful comments which considerably improved our manuscript.

Rev: “This is an interesting study with some intriguing results that the paleoceanography and sclerochronology communities would find useful. That said, the paper lacks structure and direction and the reader does ……..”

“The beginning of the introduction does not appear to be relevant based on the abstract, some very wide and irrelevant statements are made. It gets better after line 37 and reference 5. For the introduction, do not just make broad statements that are generally true but may have no relationship to your conclusions and your study………”

Author comment: We deleted lines 30 – 41 of the introduction as well as revising large parts of it by adding more general information about the investigated species. We also added more information about the purpose of this study and what we hope to achieve.

Furthermore, we also added a section about the compositional differences between the two shell layers (L354-374) as well as about the oxidative cleaning procedure (L375-387) so that the meaning is not lost in the manuscript.

We also specified that we recorded and used in-situ environmental data (L83-85)

Rev: This study has in situ temperature and salinity. This is key part of the study and makes this study unique. However, there are some key details left out. How often were the measurements taken? hourly, daily? The authors also mention ARGO data was used, but this data and the in situ data are not shown in the paper. Furthermore, the authors never discuss how time in the shells was assigned for comparison to the in situ instrumental data. This is important, and will change all the correlation results, among shells and a with shells and instrumental data.

Author comment: We added information about measurement frequency (L139-142), the depth of the landers (L137-140) and the sampled specimens (L96-99). We also prepared a new figure, that displays both the lander data and the ARGO data (Fig. 3). Time assignment was described in the original manuscript in L271-282 (now L276- 288). 

Rev: Sonication, ultrasonic cell disruptors? Were the slabs in epoxy cut more? If so how? What precautions where taken to ensure cleaning and polishing were not altering shell chemistry? Can you give the water depth the shells are from and depth of the landers recording ocean conditions?

Author comment: We added additional information about the cleaning procedure (L103-105). After embedding in epoxy the samples where not cut any further only ground and polished. Specific precautions were not taken. We are not aware of any studies that suggest chemical alteration induced by polishing or grinding. We used silicone carbide and diamond paste for grinding and polishing which should not interfere with the measurements. Considering the cleaning of the shells, we kept the time in which the samples were submerged to a minimum. Information about lander depth can be found in L137-140

Rev: The standards for LA-ICP-MS accuracy and precision: The accuracy is a large number, was the data adjust for the accuracy offsets? What Ca isotope was used for Elemental/Ca? Was the same Ca isotope also used for the internal standard? How was the ICP-MS data processed? Many labs use WaveMetrics IgorPro with Iolite software. The study only samples a small part of the shell, can the whole shell be sampled and how old does this species get?

Author comment: The LA-ICP-MS data was processed with a custom Excel sheet. The data was not corrected for the accuracy offset. The offset from reference value and RSD (Mg/Ca=5% (RSD = 3.7%), Na/Ca=6% (RSD = 5%), Sr/Ca=2% (RSD=4%) is within the same range as in other comparable studies (Marali et al., 2017b (Mg/Ca = 6%, Sr/Ca = 2%, Na/Ca = 8%)(Schöne et al., 2010 (Mg/Ca RSD = 7.2%, Sr/Ca RSD = 7.8)) where no correction is applied. 43Ca was used as both the internal standard and for normalization (L157 in the original manuscript). We also have whole shell measurements from some of the specimen but did not include them into this dataset as they do not provide much additional information within the framework of the study. Typical lifespan ranges from 50 to 80 years (López Correa et al., 2005)

Rev: Detailed items: Note some of these items (spelling, grammar, useage, etc.) are reoccurring problems in this paper. The authors should use find and replace to fix throughout the manuscript and carefully edit and revise.

Line 11 revise to “…cold-water coral reefs…”

Author comment: Corrected 

Rev: Line 13 revise to “…instrumental seawater parameters such as…”

Author comment: Corrected

Rev: Line 15-17. This is redundant, if correlation is weak, the explained variability will be low. R^2 = variability explained. Revise to be more concise and give the r value, with the n and p-values. Do this for all reported correlations throughout the paper (such as lines 18, 19). Same for sensitivity, put this value in parenthesis with its uncertainty, so the reader knows what you are referring to.

Author comment: We revised the section by adding the correlation values directly to the text or referred to an according table.

Rev: Line 20 Do you mean the correlation of elemental ratios with extension rate was the highest? The “control” is assumed, correlation is not causation. Revise.

Author comment: Revised to correlation (L24)

Rev: Line 21 The term “vital effects” is used by many authors and the definition is not clear, it could mean biological effects, reproduction effects, species effect, depth effects, calcification rates, etc. Be more specific, just say “growth-related effect”, determine by linear extension. Line 23-25 shows that these “vital effects” are quite varied.

Author comment: We changed the wording to growth related effects. We note that vital effects do not include depth effects, as vital effects describe internal biological fractionation processes (Pérez-Huerta and Andrus, 2010)

Rev: Line 22 Clarify “Multiple linear regressions” did you do regression many times or once? If once revise to “Multiple linear regression”.

Author comment: We only did the regression once per sample.

Rev: Line 22 “a large amount of elemental variability” Be specific and give and exact number.

Additionally throughout the paper, avoid using descriptive adjectives (i.e. large, low, etc.) for numerical analysis. These adjectives have no meaning, what value do you assign to “large” or “low” better to give the value and let the reader decide if the value deserve the descriptor. This removes any bias the authors place on their data.

Author comment: Thank you for that comment. We deleted descriptive adjectives wherever possible and added values or references to tables.

Rev: Line 23 Calcification rate does not equal linear extension rate. Calcification rate = linear extension rate x density. How do the authors account for density differences that would impact calcification rates? 

Author comment: We completely agree that calcification rate, growth rate, linear extension rate, etc. are often used interchangeably when they are actually not, which we also specified in the text (L516). We did not account for density differences as we do not report calcification rates and only speak of linear extension rates. We only discuss calcification rates in the context of other studies. 

Rev: Line 26-28 This needs to be explained better. Just using Mg/Sr does not mean you get around growth-related effects. Revise “combining different proxies such as Mg/Sr ratios” Proxies are some property of an archive that can be a “proxy” for some environmental parameter of interest. You mean “combining different elemental concentrations such as the ratio of Mg to Sr…” Stating Mg/Sr ratio is like saying ratio twice since it is given as a ratio with /.

Author comment: We deleted the sections about Mg/Sr as there is not enough evidence to support that conclusion.

Rev: Line 30 Revise “Intermediate water masses have been shown to respond with highly sensitivity to climate forcing”

Author comment: Deleted.

Rev: Fix throughout. Additionally, double check the “that” and “which” are used properly each time.

Author comment: Fixed.

Rev: Line 31-33. This sentence confuses me. Be more specific. What do you mean by “atmospheric climate changes” and “small scale effects” ? CO2? Water vapor, wind? Waves?If your archive-proxy is intermediate water why not just say they reflect intermediate water oceanic conditions. From your abstract, the proxies are argued to be co-varying with water temperature and salinity. What does these two properties have to do with the atmosphere?

Author comment: We deleted this section upon your recommendation.

Rev: Something is missing after “indirectly”. What does “its” refer to? Perhaps the “also” is not needed, makes more sense without also? What is being controlled? In general, avoid using “it” and its forms, be precise and tell the reader what you are referring to. Remove “it” throughout by revising the sentences.

Author comment: “Its” referred to water flow velocity. We changed the sentence to increase readability and removed the use of it throughout the manuscript.

Rev: Line 52-53 Shallow water reef corals spawn with Moon phases. I would not be surprised if cold water corals do the same, see link.springer.com/article/10.1007/BF00301193.

Author comment: Of course, there is a lot of potential triggers for coral spawning including lunar phases, salinity, temperature, food supply, solar insolation among others (Brooke and Järnegren, 2013 and references therein). Ultimately, the exact triggers are not known. In the study the reviewer referenced, the control through lunar cycles is explained to happen through the light intensity of the moon. This effect can be certainly excluded for corals that inhibit water outside the photic zone. In deep water corals, a lunar control can only be exerted through different tide cycles. The authors also state that temperature does not control the spawning activity of the observed corals because absolute temperatures are different by 5°C. It is still possible that not the absolute temperature triggers the spawning activity but the beginning decrease in temperature as the spawning period in two of the observed reefs happen right after the temperature maximum.

Rev: Line 56 The reference refers to just one cold water coral species Lophelia pertusa, and there are jundredss of these corals, so how can you extrapolate from one species to all of them? Less work has been done with cold water corals but does not mean they all do not make good archives of past conditions. This sentence is also missing a period at the end. Most of your references are for L.pertusa but there are other cold water corals that can be used for reconstructions such as bamboo corals (see doi:10.1016/j.gca.2018.12.027, doi:10.1016/j.epsl.2019.11575, doi:10.1029/2011PA002260, doi:10.1029/2010GC003443, www.sciencedirect.com/science/article/pii/S0016703711002808, Desmophyllum www.sciencedirect.com/science/article/pii/S0012821X12003652, Adkins 1998 www.sciencemag.org/cgi/content/abstract/280/5364/725, dx.doi.org/10.1016/j.gca.2011.02.019, www.sciencedirect.com/science/article/B6V66-50PVG4J-2/2/4a35c81fe923000631c5a6f4b211ef50 and others.

Author comment: Thank you for that remark. We have specifically targeted Desmophyllum pertusum for comparison, as this is the main reefbuilding coral in our sampling location (and in general). We added some of your references to show that other coral species can certainly be used for reconstructions.

Rev: Line 57-58 Reference #25 is not a coral paper but an interlab comparison paper and not an appropriate reference.

Author comment: We assume that you mean reference #22. We deleted this reference.

Rev: Line 58-59 I do not argue that bivalves are not valuable archives for reconstructions. Yet deep sea corals are still a work in progress and it is not time to give up on them. I would also argue they have simple growth patterns like a tree-rings. Additionally, bivalves tend to grow slower in colder water making seasonal reconstructions more difficult except for the intervals in the youngest parts of the shell where growth is the fastest or using laser ablation ICP methods, which can also be used with deep sea corals. As the shell ages, the seasonal resolution becomes more difficult to obtain and many researchers use annual resolution to produce long continuous reconstructions or extension rate themselves for the reconstruction. Be careful of negatives you describe for any archive-proxy because they can also impact your study!

Author comment: It was not in our interest to claim that deep sea corals are bad archives or inferior to bivalves. Just as you say research is still very much in progress and is just at the beginning. We adjusted the wording and added information about other corals that show more promising results (L54-56)

Rev: Line 61-65 What about bivalves from the tropics and subtropics? They still have growth lines and they can be annual. Give a reference that states growth lines are annual. Additionally, if you just refer to the species in your study, that would be better than generalizing for the tens of thousands of bivalves species that exist. Do all bivalves have visible growth lines and all they all annual? I bet I could find some that are not. Revise to be more specific and do not make class wide generalizations.

Author comment: We reworded the section to be more specific. L56-L58 “. A first study revealed cyclic repetitions of density changes in the shell and regular growth increment spacing, indicating a rhythmic, possibly annual control of shell deposition (López Correa et al., 2005)”

Rev: Line 65-66 This is an incomplete and confusing sentence “First observation make a seasonal accumulation of growth increments in A. excavata likely.”

Lines 66-69 Another sentence needing revision for clarity, suggestion “Different stages of growth increment development in A. excavata, which were collected at different times throughout the year, have been observed, as well as cyclic density changes in the shell, which are possibly controlled by temperature or salinity changes.

Author comment: We added a reference, deleted the generalizing statements (L54-L64) and added some more specific information about this particular species (L57-L69)

Rev: Line 70-71 Reference needed. doi.org/10.1016/j.chemgeo.2019.01.008 suggested otherwise.

Author comment: This sentence should only act as a general information. More bivalve specific information is incorporated in subsequent sentences. 

Rev: Line 78-81, For Na/Ca also see www.sciencedirect.com/science/article/B6V6R-4894W8C-89/2/9bfe9e583e5fa7fc01c95e81a6604064, and doi:10.2343/geochemj.1.0067.

Author comment: We added your suggested references

Rev: Line 86 delete “very”. This word should be avoided in technical writing unless it adds to the sentence meaning. Very is an overused word. Whenever you’re tempted to use it, try dropping it to see if any meaning is lost.

Author comment: We deleted “very” here and in multiple other locations.

Rev: Line 105 Do you mean the slab surface was polished? How were the surfaces polished?

Author comment: Yes, the slab surface was ground up to 9 µm grit with silicone carbide pads and polished up to 3 µm grit with diamond lapping paste

Rev: Line 109 and figure 1. Use of “commissure”, do you mean where the two valves meet or the shell edge? This term is usually used with brachiopods. Just say ventral margin or edge of the shell.

Author comment: We are referring to the ventral margin and changed it accordingly. Commissure is also used in context with bivalves (Bailey, 2009; Vermeij, 2013). However, we agree that the use of commissure in this particular context is wrong.

Rev: Line 100 What is “the ear of the bivalve”? Bivalves do not have ears. See Bailey, 2009: Journal of Paleontology 83, 493-495.

Author comment: We are sorry for this false translation. In german, the bivalves auricle is usually referred to as “ear”.

Rev: Line 112 “location marked in A with a dotted square” The top image is marked with an A, but there is no dotted square in in the figure. There is a dotted rectangle inside the solid rectangle, is that what you mean? The image in B looks like the orientation is from the Umbo to the ventral edge not along the growth lines. Use a A-A’ to mark the orientation and location of B.

Author comment: We added coloured lines to make the figure better understandable.

Rev: Line 113 space between number and unit, check throughout. Line 162 space between word and reference, check throughout.

Author comment: Corrected

Rev: Line 114 Revise “the opposing side”, you mean just the other side, not both sides, correct? Replace “surfaces” with “slab” since that is the term you used earlier or “slab surface”. Does this mean you cut two slabs from each shell or did you just cut them in half? I assume you have to cut thinner slabs to fit in the laser chamber.

Author comment: We cut two strips for each specimen to get matching surfaces.

Rev: Line 114 Define LA-ICP-MS at first use, it is defined in line 147. Same for all abbreviations, check throughout the paper. Line 221.

Author comment: Corrected

Rev: Line 180 Spelling mistake “recorded”. Cite the figure you are referring to.

Author comment: Corrected

Rev: Line 181 Summer and winter have more than one month, use “months”.

Author comment: Corrected

Rev: Line 180-181 Revise “Temperature varies from 7.7°C in the summer months to 8.5°C in the winter months.” The summer is colder than the winter? Odd. Is the Sula reef the same? OK, I see the explanation in lines 184-186, move this up to the temperature sentences.

Author comment: Corrected

Rev: Line 182 I do not think you mean “The Sula reef shows a temperature variability of 7°C – 8.8°C” meaning the temperature can vary by 7 to 8.8 degrees in a year but the temperature changes from 7ºC to 8.8 ºC during the year. Where are these results and data in the paper? There should be a figure and a data table for these data. If the data is already publicly available, then a citation is needed with the internet access address.

Author comment: We added figures for this data and a data table to the supplements (S6). We changed the wording for the temperature variation

Rev: Line 183 Be consistent with to or –. “33.5 to 36 g/kg in the Sula reef and 34 – 36 g/kg in...”

Author comment: Corrected

Rev: Line 191 revise “The different shell layers are not easily identifiable…” layer is more descriptive of what you are describing.

Author comment: Corrected

Rev: Line 220-223 Means are not robust against outliers. What do are the median values? 

Author comment: We added median values. While the median values are lower the difference between the two reefs is equal to the mean values.

Rev: Isn’t Fig 2 the data from the shells, not fig 3 which are scatter plots. Is the mean in line 221 for all shells or just one, clarify.

Author comment: The mean reported in line 221 is for all observed shells. 

Rev: Figure 2 I do not see the yellow line for the laser path. Perhaps use another color like blue?

Author comment: We fixed that error

Rev: Figure 3 This is a scatterplot of the elemental data for just two shells. Where is the data for the eight shells, like Figure 2? This needs to be included somewhere in the paper. There are eight shells measured, where are those plots? Establishing the annual cycle in the archive is important, as you point out in our introduction. So where is the proof of annual layers for these shells? At least have a table with the descriptive statistics for each shell, then you do not have the write it out in the results. I cannot see any “hollow” marks mentioned in the caption. How were outliers identified i.e., what statistical test did you use? How were the “fit functions” determined? With this many data points, any correlation will be significant, but not if you assess correlation with degrees of freedom. See Rodgers 1988 www.jstor.org/stable/2685263 and EBISUZAKI 1997 doi:10.1175/1520-0442(1997)010<2147:amtets>2.0.co;2.

Author comment: The data for all the measurements is in the supplements. We also added the according figures to the supplements (S7). In response to other reviewers comments we deleted the scatterplots and any connected text passages as they do not add valuable information. Like suggested we added a table with descriptive statistics

Rev: Line 226 “A correlation between Mg/Ca ratios and Na/Ca ratios is visible” where is this visible? Visible is not very scientific, give a correlation value with n and p-values. This sentences jumps to Na/Ca which has not been described yet. Talk about correlation among El/Ca after presenting results for each El/Ca.

Author comment: See former answer

Rev: Line 229 is this for all shells? See previous comments for Mg/Ca

Author comment: This is for all shells combined.

Rev: Line 230 How was no difference determined? What statistical test did you use?

Author comment: We used a regular t-test. We are aware that the t-test needs normally distributed data, which our data is not. However, the central limit theorem states for a large enough n (>100) X approximates a normal distribution (Bergström, 1949). We also tested the data with a Mann-Whitney U-test, which gave the same results.

Rev: Line 231-233 What statistical test were used to evaluate correlation? Add a correlation table for the shells and El/Ca. For an example of how this can be done see Stephens 2004 doi:10.1029/2004GL020343.

Author comment: As said earlier we deleted these sections from the manuscript.

Rev: Line 238-239 This is an interesting result, I like that you tested this. How many shells did you test this for? But where is the data and figure? Whenever you use the term “significant” there needs to be a statistical test to determine the significance. How was significance tested? Provide those details.

Author comment: Unfortunately, we only tested this for two shells. We added a data table for these results. Significance was again tested with a t-test

Rev: Line 243 Microgranular shell section Move this section up to the beginning of the results since you already started describing it. How are the growth rates different among these shells? Could growth rates be driving the differences you noted in this section of the manuscript?

Author comment: We added a table with the determined linear extension rates. Growth rate differences might play a role in the described differences but we cannot comment on that for a lack of knowledge how extension rates differ among the two shell sections. Possibly, the different crystal morphology is the driving force behind the observable differences.

Rev: Line 252 Correlating element/Ca data to instrumental data

Before jumping to correlation, where are the growth rates of the shells described? I would put all the physical properties of the shells first in the results section. Then geochemical results, cleaning results, replication results – intra and inter reef. Then you can move on to correlation between the geochemistry and environmental data.

Author comment: We revised the sections based on your suggestions.

Rev: One of the largest issue of correlating shell geochemistry to environmental data is time assignment in the archive, this is also a large source of error. Some researchers with correlate or “calibrate” the geochemistry with environmental data using just the max and min of a seasonal cycle -this is the easiest to do. Coral paleoclimatologist use Analyseries or QAnalyseries (sites.google.com/site/geokotov/software) for time assignment of many years to an SST record. Some sclerochronologists use von Bertalanffy growth curve to help with time assignments. Problem with LA-ICP-MS data is you have a lot of data that makes time assignment cumbersome but doable. Please give details on how you are assigning time to your shell data. If you assume Mg/Ca is correlating with temperature and you sue that for your time assignment, it will have the highest correlation! Please state the basis for make this assumption and test doing time assignment with the other geochemistry to see if the results vary (they should). By using Mg/Ca for time assignment, you have bias the correlation analysis. If you can use the growth lines to assign time then you can remove or reduce this bias. 

Author comment: We completely agree that the time assignment procedure is very important. In general we were using the min-max method you were mentioning with Qanalyseries. We assume that Mg/Ca is so some extent controlled by temperature, because the underlying mechanism is not controlled by an organisms biology but by thermodynamic effects (Mucci and Morse, 1990). However, we used growth increments for the majority of the time assignment. Only if the growth lines were cryptic, we utilized the Mg/Ca ratios as a further help. This method is also used in Halfar et al. 2000. 

We are not sure why utilizing other geochemical ratios should lead to different results. If we assume that Sr/Ca is similarly controlled by temperature (Stoll et al., 2002), then utilizing Sr/Ca ratios should results in the same chronology (which it does). 

Rev: Lines 257-265 Please add a correlation table to show the reader these results. The shells with low correlation, could that be a time assignment issue? Look at replication. Does the correlation improve if you build a master record with two to eight shells? The signal to noise ratio should improve as you average shells together and thus the correlations improve, if there is really a causal relationship.

Author comment: A correlation table is in the manuscript (L305). Since we only have data for 14 months we cannot use a longer Master record. The shells with low correlation can be an effect of temporal misalignment but we don’t think that is the case. Otherwise the multiple linear regression should show an equally low correlation, which is not the case.

Rev: Line 271-275 now it is mentioned that data were tuned. Shouldn’t this be mentioned before the correlation analysis? 

Author comment: The temperature and salinity correlations was not conducted with the ARGO record but the lander data. 

Rev: Line 276-282 Can you provide a citation for “nonlinearity of bivalve growth”. Does this happen within a year? Or do you mean on longer time scales? How did you “tune” the data? What software program did you use? Did you interpolate to get the geochemistry to a time scale? Include a plot of your raw shell data and your tuned shell data in your paper or appendix.

Author comment: Since we tuned our data only using the peak Mg/Ca values, we assume a constant growth rate between two fix points (which is probably not the case). We used Qanalyseries for this task without interpolating any data points. A citation for the non-linearity of shell growth would be De Ridder et al., 2004.

Rev: Line 280-283 Distance in time-geochemistry or growth lines in the shell? How many years do you have per shell? Maybe 6 from fig 2? Is this enough years to assess a relationship between extension rate and geochemistry statistically? Have you extension data per shell in your paper as well. 

Author comment: The distance was measured from growth line to growth line. We added a table with the extension data to the manuscript. In general we have 4-8 years per sample.

Rev: Table 2 How was p values assessed? # of data points or degrees of freedom? With LA-ICP-MS data there are thousands of data points so any correlation will be “significant” but you should assess degrees of freedom and re-determine the p-values, see EBISUZAKI 1997.

Author comment: We used degrees of freedom for significance tests

Rev: References need to be editing for spelling, proper capitalization for proper nouns, species names should be italics, [internet] removed and doi given instead. Access dates removed, etc.

Author comment: We corrected the references according to your suggestions

Reply to Reviewer 3

Author comment: We want to thank Reviewer 3 for his useful comments. We modified our manuscript with regards to the remarks by the reviewer.

Rev: “Abstract - - “Our results may still hold promise that A. excavata may serve as a paleoceanographic archive

for intermediate water masses despite the relatively strong vital effects, as these can potentially

be accounted for by combining different proxies such as Mg/Sr ratios.” Add a new sentence in the abstract, …more work is needed to confirm this idea.”

Author comment: We deleted this section, as we think there is not enough supporting evidence.

Rev: Line 56- missing period at end of sentence.

Author comment: Corrected

Rev: Line 58-59- some would argue that a compressing record with ontogeny is not simple.

Author comment: We agree with your comment and deleted that statement.

Rev: The study has never been replicated, either. If you are going to list one study that seemingly work for M. edulis, be sure to include other studies (e.g., Wanamaker et al., 2008) that show that the errors are very large using Mg/Ca thermometry and that vital effects seem to be dominant.

Author comment: We included your suggested study into our reference list and modified this section to make it clearer that Mg/Ca and Sr/Ca based proxies in bivalves are heavily influenced by vital effects.

Rev: Line 78- see discussion by Wanamaker and Gillikin (4.3 The use of element to calcium ratios in bivalves as environmental proxies) on this very same issue. It would be good to cite this study here. And many of the references used there are more up to date. The literature review on bivalves in this section need to be a bit more up-to-date.

Author comment: We added your suggested study to the references

Rev: Line 260. Capitalize Line.

Author comment: Corrected

Rev: Line 399- Should be Wanamaker and Gillikin

Author comment: Corrected

Reply to Reviewer 4

Author comment: The comments and remarks provided by reviewer 4 helped us greatly in improving our manuscript. We included them to the best of our knowledge and hope that this modified version fulfills your remarks.

Rev: “First, I grateful the chance to read this paper. In this study, Mg/Ca, Sr/Ca and Na/Ca ratios were measured on live-collected bivalve Acesta Excavata (Fabricius, 1779). Even though, none (or very poor) correlation has been found….”

Author comment: We have revised our manuscript and deleted the section about Mg/Sr as there is not enough evidence to support that conclusion. We also improved the way conclusions and implications are presented to make them better accessible.

Rev: “The manuscript is not very well written, and the author should be check grammatical mistakes or inconsistencies. For example: L. 56. Point after reconstruction (it is the end of the sentence). L. 260: Four with capital letter (it is the beginning of the sentence……”

Author comment: The revised the language with help of a native speaker (Co-Author D. Evans).

Rev: Figures:

- I suggest to add a new figure including a map and a picture in order to show where the shells were collected.

Author comment: We added a new figure with the sampling locations.

Rev: - Figure 2: You must include the series obtained from all shells, not only the results that you have obtained from one of them.

Author comment: We added these figure as supplement S7

Rev: - Figure 3: I guess that they have included dataset from all shells. Is that correct? If not, results obtained from all shells should be added.

Author comment: The figure included all sampled shells. We deleted the figure nonetheless as it did not add valuable information to the manuscript. It can be easily recreated from the included data.

Rev: - Figure 4: Improve this figure. The resolution is very low. Besides, add the scale.

Author comment: We improved the image quality and made the scale better visible

Rev: Specific comments:

- L. 2: Please, add "(Fabricius, 1779)". Authors should include this information in the tittle, in the abstract, as well as the first time that they write this species in the text. They also must include the complete name of the species (Acesta excavata) the first time that they cited this taxon in the abstract and in the introduction.

Author comment: We added the proper citation to the manuscript (Title, L13, L57)

Rev: - L. 13: Please, specify in the abstract what technique you used to measure this trace elements, i.e., LA-ICP-MS.

Author comment: Corrected

Rev: - L. 62-66: Authors must include more information about growth patterns of this species if this is available. How many days throughout the year this species grows? What is the longevity of this species? Information about the effect of the ontogeny in the growth patterns is available? Indicate when the sexual cycle occurs? Gametogenesis? Spawning? Is there a relationship between sexual activity and growth stoppages?

Author comment: We added more information about the bivalves biology (L57 – 69). Unfortunately, information about this species is scarce as this is one of the first studies to research the species.

Rev: - L. 90: Please, specify here what technique you used to measure these trace elements.

Author comment: Corrected (L79). We used LA-ICP-MS on an Element XR MS with an ArF Excimer Laser system from Resonetics

Rev: - L. 105: Please, explain with more details what material and methods you used for polishing shell section. Glass plates? SiC grit powder?

Author comment: We added a more thorough explanation about sample preparation. In short: We ground the samples up to 9um grid with silicone carbide sandpaper and followed with polishing up to 3um grid with diamond based lapping paste on a lapping towel.

Rev: 146: How do you know this calcium carbonate layer is calcite and not aragonite? You must investigate the shell mineral composition, for example using X-ray diffraction (XRD) or, at least, referencing a previous study where calcium carbonate composition of this species would have previously been investigated.

Author comment: We added a citation. Shell chemistry was tested by López Correa et al. 2005 with feigl solution and XRD measurments.

Rev: - L. 181: Why do you have warmer temperatures in winter? This is a little strange for me and maybe for other readers too. Please, try to explain this. I think that it is very normal and of course that there is a very simple explanation, but it is strange.

Author comment: We can understand the confusion. An explanation was given in L184-186 in the orginial document (now L186). This is caused by the replacement of the cold and brakish Norwegian coastal current (NCC) by the North Atlantic current (NAC), which is warmer and saltier. During summer south-westerly winds induce a downwelling of the NCC, leading to cooler temperatures in the bottom water. During winter and spring, northerly wind and icnraesed meltwater input displace the NCC upwards, which allows the warmer NAC to replace the bottom water (Jacobson, 1983; Sætre and Ljøen, 1971; Sakshaug and Myklestad, 1973)

Rev: - L. 204: The same that previously. How do you know that layers are composed by aragonite or calcite? Please explain that.

Author comment: We added a citation. Shell chemistry was tested by López Correa et al. 2005 with feigel solution and XRD measurements.

Rev: - L. 205-206: “These lines coincide with minima in Mg/Ca ratios, although these lines are not always visible”. The sentence is very tricky for a good understanding. How do you know that minima Mg/Ca ratios systematically match with these lines if they are not always visible? Please, you should explain this better. Why should I suppose then that it is true if you cannot see all lines? That is crucial.

Author comment: We agree that the chronology is a crucial step. In the majority of cases we are able to identify the growth increments. In the cases we did not success, we tried every available technique (mutvei solution, fluorescence microscopy, confocal raman spectroscopy) without success. In the cases where we did not observe growth lines in the thin sections, we did still observe the on the outer shell surface.

Rev: - L. 226: I guess that these results about correlations between trace-element were extracted from figure 3. So, they should reference it in the text.

Author comment: That’s correct. However, we think this figure does not add valuable information and deleted it.

Reply to Reviewer 5

Rev: This manuscript’s dataset has potential to make an interesting contribution to investigations of mollusc shell elemental ratios as proxies for reconstruction of seawater parameters (i.e. temperature and salinity).

The key strength of the study is the ability to make comparisons between mollusc shell chemical proxy records and instrumental records of seawater temperature and salinity, as well as current velocity. However, these important oceanographic data are not presented within the manuscript and should be included as separate panels on a new figure. Figure 5 also should be expanded to include time-series analysis of the instrumental seawater temperature and salinity datasets (to complement the current velocity assessment), even if no clear periodicity is event. Without sight of the oceanographic data it is not possible to discern if the shell Mg/Ca records are even tracking seawater temperature. The manuscript also should be explicit with regards to the distance between the lander oceanographic instrumentation and the sampled mollusc specimens, and it is rather unclear why ARGO data are preferred to the available lander instrumentation. A site map, to support text lines 95 to 98, also would be useful.

Author comment: We added the lander recorded data for temperature and salinity of the two reefs as well as spectral analysis data and a Map of the sampling locations. 

We did not prefer the ARGO data. Correlations between E/Ca data and environmental data was calculated using the Lander data. ARGO data is only used because it offers a longer temperature record which we used to asses if our chronology based on growth increments fits this record.

Rev: The key weakness with the manuscript is the approach taken to convert distance along shell growth axes to time, i.e. the clearly demonstrated very weak relationship between shell Mg/Ca and seawater temperature really does not support use of Mg/Ca minima as tie points for age model development, and how interpolation between such points has been undertaken is rather unclear (i.e. constant or variable growth rates). Furthermore, it really does seem to be quite a circular argument to develop an age model using one elemental ratio (i.e. Mg/Ca) and then test for a relationship between that same proxy and seawater temperature. Also, looking at Figure 2 the ‘peak’ in Mg/Ca seems to occur in different positions relative to the ‘assumed yearly growth lines’, seemingly further negating the age model approach that has been adopted. (The yellow dashed line on Figure 2 also is not clear.) Finally, the cross section of the shell shown in Figure 4 apparently shows a clear discontinuity in shell growth, presumably due to a significant external forcing rather than a response to a slower change to external seawater conditions; could such an abrupt forcing have been changing current speed (data are not plotted to assess) or even predation? Cross sections of the other investigated mollusc specimens, even if only included within a supplementary information section, would be useful for comparative purposes and enable a more complete assessment of shell growth variability between the studied specimens.

Author comment: The reviewer is right about the discontinuity in Fig 4. In this shell section a small rock was embedded in the shell. The ventral margin of these bivalves is coarse which makes it easy for small particles to get stuck. We guess one of this particles was simply overgrown by shell material. This feature almost certainly represents some kind of strong external forcing. 

The age model was not developed on the basis of the Mg/Ca measurements only if the growth increments were cryptic we relied on Mg/Ca measurements. This approach was also used on coralline red algae (Halfar et al., 2000, 2008). In general we used growth lines on the shell surface and shell thin sections.

Rev: The text on lines 316 to 333 implies that an age model can be developed using the internal shell increment growth series (as shown in the bottom right panel of Figure 2), such an age model then being entirely independent of the shell elemental ratio proxies and thereby facilitating considerably more robust assessment of any relationships between shell chemistry and seawater parameters. This section, which should be called sclerochronology (and not scleromorphology) should appear after presentation of the oceanographic data and before any discussion of the mollusc shell elemental contents. It should also be possible to include graphs that present a comparison of the shell growth increment sizes (and thus growth) between the individual specimens investigated, to discuss further the comparability between the different shells for the same time period, following presentation of a robust argument for the factor(s) controlling shell increments, e.g. tidal periodicity as is mentioned within the manuscript’s text. Of course, an alternative approach to support development of an independent age model would be via use of shell oxygen-isotope ratios which could then be compared to predicted values derived from the measured seawater temperature and salinity datasets (not many studies are in a position to adopt such an approach, which this dataset actually does facilitate); such an approach also would help resolve the question as to whether there is any seasonal cessation of shell growth in the studied locations (theoretically also identifiable by counting of growth increments between growth lines, if the latter are not formed by irregular events such as predation). Mark and recapture of mollusc shells also would have helped with robust development of age models, but such an activity presumably was not completed in this study?

Author comment: We added a table with the yearly growth increments for every investigated sample (Table 7). The age model was constructed using visible growth lines on the shell surface and shell thin sections. 

Unfortunately, we did not mark and recapture the specimens. We added a table with the growth increment size.

Rev: The value of Figure 3 is very unclear, because the primary interest is whether the shell elemental ratios relate to oceanographic parameters, rather than to each other. Furthermore, other studies have shown that different mollusc species can exhibit differences in elemental ratios between individuals (cultured in the same experimental aquaria), such that merging all specimens together potentially hides inter-specimen variability. In fact, the caption to Figure 3 confirms this to be the case for this dataset, because two specimens are excluded as ’outliers’! If something like Figure 3 is to be presented then there should also be separate panels for each individual specimen. What would be much more useful would be to replicate Figure 2 for each of the other seven specimens, ideally plotted against an independent growth-increment (or oxygen-isotope ratio) derived age model to test whether multiple specimens from the same location(s) faithfully record consistent elemental ratios through time. Only one specimen is shown in detail (in Figure 2) at present, and plotting of elemental ratios for all individuals against time also would facilitate testing of whether the shell chemistry better follows ambient oceanographic forcing at certain times of the year, which cannot be discerned when the data are lumped together as is the case in the current manuscript. Note: a supplementary information file can be used to include multiple figures, if they will not fit within the main article.

Author comment: We agree with the reviewers comment about figure 3. We deleted the figure because we do not think it adds valuable information. We also included additional figures to the supplement material which show E/Ca data against the chronology for all investigated species. 

Rev: Overall, there is likely considerably more variability and interest within the dataset than is discussed at present. For example, once independent age models are developed, Figure 7 should also plot the relationships for individual specimens and not just all specimens merged together for the two reefs. Statistical testing also could be used to test for differences between specimens, and between the two sampled reefs. Furthermore, from Table 1 it is evident that all eight specimens investigated were of a comparable size (ca. 100 mm in length) and thus all adults. Could data also be included for some smaller, i.e. juvenile, individuals, so that an assessment could be made as to whether ontogeny has any influence on elemental ratios within this species? Another more minor point, was any shell periostracum removed prior to determination of shell organic content? And, can parts of Figure 4 be presented at higher magnification?

Author comment: We picked the specimens to be of roughly equal size. Enough smaller specimens were not available for comparison. However, we also measured E/Ca ratios over the complete life of some specimens which did not show any ontogenetic influence.

We did not specifically clean the periostracum prior to the organic content determination. We only cleaned the shell in an ultrasonic bath for ca. 5 minutes.

Rev: Finally, the current Discussion section is not that coherent; there are long paragraphs that contain multiple short statements that are not very well interlinked. Some statements also seem rather too speculative, such as in relation to selective dissolution of calcium from a mollusc shell? Ultimately, the relationships between elemental ratios and oceanographic parameters (i.e. temperature and salinity) are very weak and thus of limited utility. The manuscript tends to overstate what might be possible, e.g. arguing that Mg/Sr ratios are better than Mg/Ca, but the r-squared values (for all data combined) only improve from 0.07 to 0.15 (from Figure 7)! There could well be more significant relationships embedded within this dataset, but such is not evident from the approach that was taken to merge all data values together.

Author comment: We revised large parts of the manuscript to increase readability. Sections that we deemed overly speculative were removed. We also deleted parts that “overstate what might be possible”. This includes sections about Mg/Sr as well as the mentioned section about selective dissolution.

---

## [Decision Letter · Decision Letter 1]

27 Oct 2020

PONE-D-20-12353R1

Compositional variability of Mg/Ca, Sr/Ca, and Na/Ca ratios in the deep-sea bivalve *Acesta excavata*

PLOS ONE

Dear Dr. Schleinkofer,

Thank you for submitting your manuscript to PLOS ONE. After careful consideration, we feel that it has merit but does not fully meet PLOS ONE’s publication criteria as it currently stands. Therefore, we invite you to submit a revised version of the manuscript that addresses the points raised during the review process.

Please pay close attention to the reviewers comments in your revision. All 3 reviewers found the manuscript to be improved, but 2 still ask for some significant work. Notably, the reviewers recommend different decisions ranging from minor revision-reject, so I would strongly urge you to make sure you address the points made by all reviewers in this revision. Two reviewers ask for improvements to the statistics are requested, and possible improvement of the implications of the study. I look forward to receiving your revised manuscript in due course.

We look forward to receiving your revised manuscript.

Kind regards,

Tim M. Conway, Ph.D.

Academic Editor

PLOS ONE

Additional Editor Comments (if provided):

I have now received three reviews of your re-submission of this manuscript, which are attached. As you will see, Reviewer 2 and 3 find the paper improved, but still suggest significant work may be needed before this paper can be considered publication-ready. Please revise your manuscript in response to these review comments and take close notice of all their comments. Please especially pay attention to the comments which mention carefully wording the implications of your study from such a small number of samples.

Reviewers' comments:

Reviewer's Responses to Questions

**Comments to the Author**

1. If the authors have adequately addressed your comments raised in a previous round of review and you feel that this manuscript is now acceptable for publication, you may indicate that here to bypass the “Comments to the Author” section, enter your conflict of interest statement in the “Confidential to Editor” section, and submit your "Accept" recommendation.

Reviewer #2: (No Response)

Reviewer #3: All comments have been addressed

Reviewer #4: (No Response)

2. Is the manuscript technically sound, and do the data support the conclusions?

Reviewer #2: No

Reviewer #3: Yes

Reviewer #4: Partly

3. Has the statistical analysis been performed appropriately and rigorously? 

Reviewer #2: No

Reviewer #3: Yes

Reviewer #4: Yes

4. Have the authors made all data underlying the findings in their manuscript fully available?

Reviewer #2: Yes

Reviewer #3: Yes

Reviewer #4: Yes

5. Is the manuscript presented in an intelligible fashion and written in standard English?

Reviewer #2: No

Reviewer #3: Yes

Reviewer #4: Yes

6. Review Comments to the Author

Reviewer #2: Overall the paper is improved but still needs work to be publishable. The statistical aspects of the paper need to be addressed better but should not change the overall results.

General observations:

I suspect growth-related effects are a source of the problem with these bivalves, have the authors separated out faster and slower growing shells and compare their chemistry that way? The growth-related effect on biological carbonate material have been explored in corals, first documented by McConnaughey, T. (1989) for δ18O and deVilliers et al. (1994) for Sr/Ca. See Kuffner et al. (2017) figure 3 for an example of how to make this assessment.

The study sample the outer increments or younger part of the shell that is slower growing thus, slower growth could be driving El/Ca. Why didn’t the authors sample the older faster growing portions of the shell? I understand in tree-ring studies that the early years of growth have a growth trend that is removed for tree-ring increment studies. However, shells are not trees and this study is looking at chemistry not growth increments. I would think you would sample the initial growth years when growth is faster to avoid the slower growth- related effects. This should eb explained in the introduction and study design and then discussed in the discussion section.

Next, is replication, I assume the shells are all collected at the same time so the authors should be able to assess reproducibility of the El/Ca signals among the shells from each location and then if there is any correlation between locations. If the shells have a same signal, then the forcing is environmental, if not biological. This is not explored in the statistical tests of the study.

Correlation must be reported with the n used for the significance testing. Correlations < 0.1 are dubious regardless of the p-valve because “n” is degrees of freedom in the correlation equation, not number of observations. Laser Ablation sampling generates a lot of data; thus, I suspect the n values are high (>1000) and thus why the authors’ p values are low suggesting significance. The authors data has serial correlation in it, and this MUST be considered in the evaluation of correlation significance determine with degrees of freedom, NOT the number of observations. Looking at Figure 5 there is 6 years of data, so the authors’ degrees of freedom would be close to 12. However, if the authors determine the degrees of freedoms, the authors will find these correlation results are probably different and possibly not significant. The authors are can use the Runs test or best method is Ebisuzaki, W. (1997). "A Method to Estimate the Statistical Significance of a Correlation When the Data Are Serially Correlated." Journal of Climate 10(9): 2147-2153. If the correlations are weak and the authors are not going to show the scatterplots in the main paper, then why mention the correlation and their details in the abstract? Same for the T-test and f-test (in Table 3), the n should be degrees of freedom. With more than 1000 data points almost any r will be significant (r >0.062 is significant at 5% level). See Zwiers, F. W. and H. von Storch (1995). "Taking serial correlation into account in tests of the mean." Journal of Climate 8: 336-351.

Other statistical test questions that are not clear:

• Table 2, 3, 4 How many annual cycles or years were compared for these shells, i.e. are the authors comparing the same time intervals or varying time intervals for all the shells. Looking a Fig S7, the data spans different number of years in the shells. The same interval and same amount of data of time should be made for the comparison tests (correlation, t-test, f-test, etc.).

Line 270 Table 4 reports means, and the T-test is for means only. Revise “Significant mean differences are observable for every E/Ca in all measured samples.”

• Explain how the multiple linear regression was performed, give citation or software used.

• Add a table that gives the time interval each shell spans. Explain what interval for each shell was used for statistical tests. For a robust statistical assessment, the same time interval should be assessed for each shell not 5 years in one and 2 years in another. Looking at Table 7 the time interval for each shell varies and the common interval is just two years when excluding 6R. The authors could expand the windows and see how statistical tests vary with more years but less shells.

• The statistical results and text in the abstract are distracting (i.e., r=0.05, p<0.01), just tell the reader what you found.

•was the cleaning test and organic matter tests for whole shell or a similar interval as the LA-ICP-MS data? See previous comments about time intervals in shells and statistical tests.

Authors should carefully proof reader for typos, spelling mistakes, punctuation, and figure and tables numbers are correct.

The authors have made their data available in the publication but they should also make it available in the community accepted repositories at NOAA Paleoclimatology (Khider et al. 2019) https://www.ncdc.noaa.gov/data-access/paleoclimatology-data or Pangaea https://www.pangaea.de/.

Line 271 Same as previous comment, “Mean Mg/Ca are lower in the microgranular shell layer.” Fix elsewhere. Looking at Fig. 5 the Mg/Ca are not always lower in the microgranular shell layer.

Line 269-278 The authors should also reference and discuss the plots of the data in the Fig. 5 and Fig S7.

Specific items to address:

Line 17 revise for plural agreement “parameters”.

Abstract and elsewhere. Correlations must be reported with the degrees of freedom used to determine p-values. Where these for the Fibrous or Microgranular parts of the shell? Revise abstract to be more informative to the reader. It is OK to report negative results, these are important so others do not repeat your negative findings.

Line 63 Revise for missing verb “…first phase lasts until the bivalve has built 18-22 increments…” Does an increment – one year? If so, explain this.

Line 64 “crammed” means to be “completely fill” is a not a good word to use here, use a more scientifically descriptive word like “tightly aligned”. Delete “much” it is not needed.

Line 78-79 This is not a complete sentence, revise or use correct punctuation.

Line 83 The elemental ratios are written as a ratio Mg/Ca, do not say “ratio” afterwards, it is redundant. The “/” means divides by or ratio. Fix throughout the paper.

Line 114 “macroscopically visible” is redundant macroscopic means visible without magnification and it is an adjective, macroscopically is not a word.

Line 133 “The data shown here” Where are these data the authors say are shown? I think the authors mean “the data used”.

Methods section grammar needs to be addressed. The first section “Sampling” is written in past tense and the next section “Oceanographic data” starts in present tense and then switches to past tense. Use one tense (past) throughout the methods section.

Line 155 Fig. 1 is the study site map, the authors mean Fig. 2B.

Line 192 Need space between fig. and number. Fix elsewhere (line 227).

Line 214 in correct use of “/” in “aragonite/calcite” replace with a hyphen.

The captions and figure # seem to not correspond for Fig 4 and 5. The caption for fig. 5 seems to be to be for the figure labeled fig 5 and vice versa. Line 215 refers to fig 4 but the caption for fig 4 does not make sense, it looks to be referring to the fig labeled fig. 5.

Line 215 What do the authors mean by “visible sculpture”? the jagged edge? A “sculpture” is a work of art of in two- or three-dimensional representative or abstract forms. I do not think this is the right word.

Line 252 What exactly do the authors mean by “There is no resolvable difference in Sr/Ca ratios between the two sample sites”? Did the authors do a T-test for means and F-test for variances? Do they look like they plot of top of each other the same? If so, show this. Do they co-vary , do the growth intervals appear to be the same. Explain this better with no detail.

Line 282 Revise “…own observations, we assume the growth lines were produced on an annual basis.”

Line 289 Revise “…temperature, as well as Mg/Ca,…” Spelling mistake

Line 303 period is missing at end of sentence.

Line 358-363 The authors should be cautious in their interpretation. They tested mean values between the shell layers. Looking at Fig. 5 the two layers appear to track each other in all three e/Ca. A correlation between the layers how help make this assessment.

Figure 4 caption and Figure 2 Explain and show what the authors mean by the “commissure line on the shell”.

Table 1 there is “229” in the bottom right cell, not sure that is supposed to be there.

Supplemental Figure 7 is too small to read the values and look at the shells, please put these on multiple pages and enlarge images figure Fig. 5. Same for line 254. Make the scale for El/Ca for each shell the same so the reader can compare visually. These figures should show microgranular and fibrous data.

Table 4 How many annual cycles or years were compared for these shells, i.e. are the authors comparing the same time intervals or varying time intervals for all the shells. Looking a Fig S7, the data spans different number of years in the shells. The same interval and same amount of data of time should be made for the comparison tests.

Table 5 Explain in the caption. Explain in the caption how All samples was determined, the same time intervals, or just all the data produced? If different intervals, then sample size needs to be considered.

Table 6 Explain in results or in the methods how the multiple linear regression was performed, give citation or software used.

Line 389 Spelling of “therefor” is incorrect.

Line 398 What is the spatial-temporal resolution of the authors’ study? Add this to the table with the time interval present in the data for each shell that I requested.

Line 532 Revise to be more specific, “…A. excavata are not well suited for paleo temperature and salinity reconstructions due to weak correlations between proxy trace elements and these environmental variables,…” Same for Line 547

Conclusions- I like the discussion and conclusions- use these to revise the abstract.

References cited:

de Villiers, S., Shen, G.T., Nelson, B.K., 1994. The Sr/Ca-temperature relationship in coralline aragonite: Influence of variability in (Sr/Ca)seawater and skeletal growth parameters. Geochimica et Cosmochimica Acta, 58(1): 197-208, doi:10.1016/0016-7037(94)90457-x.

Khider, D., Emile‐Geay, J., McKay, N.P., Gil, Y., Garijo, D., Ratnakar, V., Alonso‐Garcia, M., Bertrand, S., Bothe, O., Brewer, P., Bunn, A., Chevalier, M., Comas‐Bru, L., Csank, A., Dassié, E., DeLong, K., Felis, T., Francus, P., Frappier, A., Gray, W., Goring, S., Jonkers, L., Kahle, M., Kaufman, D., Kehrwald, N.M., Martrat, B., McGregor, H., Richey, J., Schmittner, A., Scroxton, N., Sutherland, E., Thirumalai, K., Allen, K., Arnaud, F., Axford, Y., Barrows, T.T., Bazin, L., Pilaar Birch, S.E., Bradley, E., Bregy, J., Capron, E., Cartapanis, O., Chiang, H.W., Cobb, K., Debret, M., Dommain, R., Du, J., Dyez, K., Emerick, S., Erb, M.P., Falster, G., Finsinger, W., Fortier, D., Gauthier, N., George, S., Grimm, E., Hertzberg, J., Hibbert, F., Hillman, A., Hobbs, W., Huber, M., Hughes, A.L.C., Jaccard, S., Ruan, J., Kienast, M., Konecky, B., Le Roux, G., Lyubchich, V., Novello, V.F., Olaka, L., Partin, J.W., Pearce, C., Phipps, S.J., Pignol, C., Piotrowska, N., Poli, M.S., Prokopenko, A., Schwanck, F., Stepanek, C., Swann, G.E.A., Telford, R., Thomas, E., Thomas, Z., Truebe, S., Gunten, L., Waite, A., Weitzel, N., Wilhelm, B., Williams, J., Williams, J.J., Winstrup, M., Zhao, N., Zhou, Y., 2019. PaCTS 1.0: A Crowdsourced Reporting Standard for Paleoclimate Data. Paleoceanography and Paleoclimatology, 34(10): 1570-1596, 10.1029/2019pa003632.

Kuffner, I.B., Roberts, K.E., Flannery, J.A., Morrison, J.M., Richey, J.N., 2017. Fidelity of the Sr/Ca proxy in recording ocean temperature in the western Atlantic coral Siderastrea siderea. Geochemistry, Geophysics, Geosystems, 18(1): 178-188, 10.1002/2016GC006640.

McConnaughey, T., 1989. 13C and 18O isotopic disequilibrium in biological carbonates: II. In vitro simulation of kinetic isotope effects. Geochimica et Cosmochimica Acta, 53(1): 163-171,

Reviewer #3: The authors addressed my concerns from the previous revisions. The data and the study will be useful for other researchers and now that the others have more carefully assessed the physiological/vital effects that obscure the recorded environmental signals in the shell elemental ratios.

Reviewer #4: I grateful to editor the opportunity to review this manuscript. After reading it, unfortunately I should recommend its rejection. They have done a very good job and the methodology seems very strong. However, I am not very confident in the value of this manuscript for a very high-quality journal like Plos One.

The study conducted by the authors seems based on a very strong-methodology, but the results obtained have not implications for future climate studies. Please, take into account that it is not my subjective opinion, authors are who conclude the manuscript writing: "it can be stated that Mg/Ca, Sr/Ca and Na/Ca ratios in A. excavata are unlikely to be good proxies for environmental reconstructions at this point". They also admit it in the discussion section: “…these results indicate that Mg/Ca, Sr/Ca and Na/Ca are unlikely to be used for environmental reconstruction”. I agree, these E/Ca ratios are not reliable climate proxies. In four shells, they found a correlation lower than 0.1 between SST and Mg/Ca ratios. Mean correlations between E/Ca ratios and SST or Salinity are 0.06, 0.02, 0.17, 0.02, 0.04 and 0.12. I am so sorry, but these coefficients are not reporting a weak correlation as authors said, they are actually showing an absent of correlation. In this same sense, the authors are who admit that only 6% of the variability of Mg/Ca ratios can be explained by SST changes through mollusk life span. It is not possible to say at same time that 0.06 is a weak correlation while then they say that magnesium precipitation is only driven by SST in a 6%. In summary, manuscript have valuable data, a strong methodology, but the results, conclusions and specially its implications for future studies are not good enough for a very high impact journal like this.

In any case, if editor decide to admit it or if the other reviewers have favorable opinions, please check the manuscript again very carefully, there are several mistakes. Some examples: L. 155: Fig. 1 should be Fig. 2. L. 287. You said that maximum correlation value is 0.64, but in the table 5 the highest value is 0.63. L. 389. Please, correctly write “therefore”.

7. PLOS authors have the option to publish the peer review history of their article (what does this mean?). If published, this will include your full peer review and any attached files.

Reviewer #2: No

Reviewer #3: No

Reviewer #4: No

---

## [Author Response · Author response to Decision Letter 1]

26 Nov 2020

Reviewer #2: Overall the paper is improved but still needs work to be publishable. The statistical aspects of the paper need to be addressed better but should not change the overall results.

General observations:

I suspect growth-related effects are a source of the problem with these bivalves, have the authors separated out faster and slower growing shells and compare their chemistry that way? The growth-related effect on biological carbonate material have been explored in corals, first documented by McConnaughey, T. (1989) for δ18O and deVilliers et al. (1994) for Sr/Ca. See Kuffner et al. (2017) figure 3 for an example of how to make this assessment.

We thank the reviewer for that valuable comment.

We did pick the investigated shells to be of roughly equal size to ensure they are in a comparable life cycle. We did the assessment you suggested (based on Kuffner et al. 2017) but we do not observe a regular trend of the residuals as in your suggested paper. However, since we only report annual extension rates and our temperature regression only spans one year we cannot exactly recreate the method by Kuffner et al. 2017

Rev. 2 : The study sample the outer increments or younger part of the shell that is slower growing thus, slower growth could be driving El/Ca. Why didn’t the authors sample the older faster growing portions of the shell? I understand in tree-ring studies that the early years of growth have a growth trend that is removed for tree-ring increment studies. However, shells are not trees and this study is looking at chemistry not growth increments. I would think you would sample the initial growth years when growth is faster to avoid the slower growth- related effects. This should eb explained in the introduction and study design and then discussed in the discussion section.

The main reason we are sampling the younger shell parts is because we only have environmental data for 07/2013 – 07/2014. For samples that were picked alive in 07/2014 we can be sure that the last growth increments relates to the recorded environmental data.

In addition, inorganic precipitation experiments show that, for Mg and Sr, the calcification process happens in equilibrium with the calcification fluid at low calcification rates [1]. Therefore, its preferable to use slow growing shell areas instead of fast growing areas.

Rev. 2:Next, is replication, I assume the shells are all collected at the same time so the authors should be able to assess reproducibility of the El/Ca signals among the shells from each location and then if there is any correlation between locations. If the shells have a same signal, then the forcing is environmental, if not biological. This is not explored in the statistical tests of the study.

All shells were collected in summer 2014. We had this data in a previous version but deleted upon suggestion of another reviewer. We added another figure to the supplementary material that shows the intra-reef variability of the E/Ca ratios.

Rev. 2:Correlation must be reported with the n used for the significance testing. Correlations < 0.1 are dubious regardless of the p-valve because “n” is degrees of freedom in the correlation equation, not number of observations. Laser Ablation sampling generates a lot of data; thus, I suspect the n values are high (>1000) and thus why the authors’ p values are low suggesting significance. The authors data has serial correlation in it, and this MUST be considered in the evaluation of correlation significance determine with degrees of freedom, NOT the number of observations. Looking at Figure 5 there is 6 years of data, so the authors’ degrees of freedom would be close to 12. However, if the authors determine the degrees of freedoms, the authors will find these correlation results are probably different and possibly not significant. The authors are can use the Runs test or best method is Ebisuzaki, W. (1997). "A Method to Estimate the Statistical Significance of a Correlation When the Data Are Serially Correlated." Journal of Climate 10(9): 2147-2153. If the correlations are weak and the authors are not going to show the scatterplots in the main paper, then why mention the correlation and their details in the abstract? Same for the T-test and f-test (in Table 3), the n should be degrees of freedom. With more than 1000 data points almost any r will be significant (r >0.062 is significant at 5% level). See Zwiers, F. W. and H. von Storch (1995). "Taking serial correlation into account in tests of the mean." Journal of Climate 8: 336-351.

You are right that the total data for each sample is serially correlated. However, the data we used to calculate the correlation is not because it only includes one annual cycle (see S5). We added degrees of freedom to the tables (L268, L277,L307, L313)

Other statistical test questions that are not clear:

Rev. 2: Table 2, 3, 4 How many annual cycles or years were compared for these shells, i.e. are the authors comparing the same time intervals or varying time intervals for all the shells. Looking a Fig S7, the data spans different number of years in the shells. The same interval and same amount of data of time should be made for the comparison tests (correlation, t-test, f-test, etc.).

In Table 2 and 3 we used all available data. In Table 4 we used only the last annual cycle. We redid the analysis with the same time interval for all shells and found no differences in the results.

Rev. 2:Line 270 Table 4 reports means, and the T-test is for means only. Revise “Significant mean differences are observable for every E/Ca in all measured samples.”

Corrected L280

Rev. 2: Explain how the multiple linear regression was performed, give citation or software used.

We added a section about the statistical analysis, which program we used and the used tests. L190

“All statistical calculations were conducted with OriginPro 2020. We used the T-test to compare means of different populations and a linear regression model to investigate the relationship between predictor and response variables….”

Rev. 2:Add a table that gives the time interval each shell spans. Explain what interval for each shell was used for statistical tests. For a robust statistical assessment, the same time interval should be assessed for each shell not 5 years in one and 2 years in another. Looking at Table 7 the time interval for each shell varies and the common interval is just two years when excluding 6R. The authors could expand the windows and see how statistical tests vary with more years but less shells.

The time interval of each sampled shell is given in table 7. For the correlation between E/Ca and Salinity-temperature we used the same time interval for each shell (1 year). 

Rev. 2:The statistical results and text in the abstract are distracting (i.e., r=0.05, p<0.01), just tell the reader what you found.

We added the statistical results upon suggestion of another reviewer. However, we agree that it is distracting.

Rev. 2: was the cleaning test and organic matter tests for whole shell or a similar interval as the LA-ICP-MS data? See previous comments about time intervals in shells and statistical tests.

We used the same interval for the cleaning test as for the LA-ICP-MS data. 

Rev. 2:Authors should carefully proof reader for typos, spelling mistakes, punctuation, and figure and tables numbers are correct.

We carefully eliminated occurring typos and wrong table – figure numeration.

Rev. 2:The authors have made their data available in the publication but they should also make it available in the community accepted repositories at NOAA Paleoclimatology (Khider et al. 2019) https://www.ncdc.noaa.gov/data-access/paleoclimatology-data or Pangaea https://www.pangaea.de/.

We are currently preparing the data to be uploaded to Pangaea

Rev. 2:Line 271 Same as previous comment, “Mean Mg/Ca are lower in the microgranular shell layer.” Fix elsewhere. Looking at Fig. 5 the Mg/Ca are not always lower in the microgranular shell layer.

Corrected to mean Mg/Ca L281

Rev. 2:Line 269-278 The authors should also reference and discuss the plots of the data in the Fig. 5 and Fig S7.

We added references to these figures.

Specific items to address:

Rev. 2: Line 17 revise for plural agreement “parameters”.

Corrected. L17

Rev. 2: Abstract and elsewhere. Correlations must be reported with the degrees of freedom used to determine p-values. Where these for the Fibrous or Microgranular parts of the shell? Revise abstract to be more informative to the reader. It is OK to report negative results, these are important so others do not repeat your negative findings.

We revised the abstract according to your suggestion and added DF to reported correlations in the tables.

Rev. 2:Line 63 Revise for missing verb “…first phase lasts until the bivalve has built 18-22 increments…” Does an increment – one year? If so, explain this.

Yes, 18 increments equal 18 years [2]. We made the statement clearer. L66

Rev. 2:Line 64 “crammed” means to be “completely fill” is a not a good word to use here, use a more scientifically descriptive word like “tightly aligned”. Delete “much” it is not needed.

Corrected. L68

Rev. 2:Line 78-79 This is not a complete sentence, revise or use correct punctuation.

Corrected.

“Whether this assumption holds true for other species has to be further investigated.L81

Rev. 2:Line 83 The elemental ratios are written as a ratio Mg/Ca, do not say “ratio” afterwards, it is redundant. The “/” means divides by or ratio. Fix throughout the paper.

You are right that adding the word “ratio” might be redundant, however this redundancy is often used in PLOS ONE articles.

“Whether using the δ18O signatures or Mg/Ca ratios from foraminiferal tests…” [3]

“ The molar magnesium/calcium (Mg/Ca) ratio (mMg/Ca) of seawater has varied between 0.5 and 5.3 through the current Phanerozoic eon [1].”[4].

We adjusted the wording in some instances but we think that it increases the readability and makes some statements better understandable.

Rev. 2:Line 114 “macroscopically visible” is redundant macroscopic means visible without magnification and it is an adjective, macroscopically is not a word.

We want to object that the term „macroscopically visible“ is commonly used in PLOS ONE articles in the same sense we used it, being that the features are visible with the bare eye.. 

„The macroscopically visible appearance of the satellite cluster is indicated with a red circle.“[5]

“The extent of macroscopically visible cell density observed across” [5]

“…, no macroscopically visible alterations of the skin were found 2.5 weeks after infection (Fig. 1, A1 and A2).”[6]

Rev. 2:Line 133 “The data shown here” Where are these data the authors say are shown? I think the authors mean “the data used”.

Corrected. L136

Rev. 2: Methods section grammar needs to be addressed. The first section “Sampling” is written in past tense and the next section “Oceanographic data” starts in present tense and then switches to past tense. Use one tense (past) throughout the methods section.

We adjusted the wording to have a regular tense.

Rev. 2: Line 155 Fig. 1 is the study site map, the authors mean Fig. 2B.

Corrected. L158

Rev. 2: Line 192 Need space between fig. and number. Fix elsewhere (line 227).

Corrected

Rev. 2: Line 214 in correct use of “/” in “aragonite/calcite” replace with a hyphen.

Corrected. L223

Rev. 2: The captions and figure # seem to not correspond for Fig 4 and 5. The caption for fig. 5 seems to be to be for the figure labeled fig 5 and vice versa. Line 215 refers to fig 4 but the caption for fig 4 does not make sense, it looks to be referring to the fig labeled fig. 5.

Corrected

Rev. 2: Line 215 What do the authors mean by “visible sculpture”? the jagged edge? A “sculpture” is a work of art of in two- or three-dimensional representative or abstract forms. I do not think this is the right word.

Corrected to jagged edge.L224

Rev. 2: Line 252 What exactly do the authors mean by “There is no resolvable difference in Sr/Ca ratios between the two sample sites”? Did the authors do a T-test for means and F-test for variances? Do they look like they plot of top of each other the same? If so, show this. Do they co-vary , do the growth intervals appear to be the same. Explain this better with no detail.

We want to express here that based on a T-test there is no difference in the mean Sr/Ca ratio between the two reefs. However, we deleted the statement because it adds no necessary information.

Rev. 2: Line 282 Revise “…own observations, we assume the growth lines were produced on an annual basis.”

Corrected. L292

Rev. 2: Line 289 Revise “…temperature, as well as Mg/Ca,…” Spelling mistake

Corrected

Rev. 2: Line 303 period is missing at end of sentence.

Corrected

Rev. 2: Line 358-363 The authors should be cautious in their interpretation. They tested mean values between the shell layers. Looking at Fig. 5 the two layers appear to track each other in all three e/Ca. A correlation between the layers how help make this assessment.

We are not entirely sure what the reviewer is suggesting here. Indeed, the two layers track each other in every measured E/Ca ratio but the mean ratios are different. These differences are most likely caused by the different crystal morphology in the layers. 

Rev. 2: Figure 4 caption and Figure 2 Explain and show what the authors mean by the “commissure line on the shell”.

In Figure 4 we mean the ventral margin of the shell. We changed the wording to be clearer. We cannot find the term in Figure 2.

Rev. 2: Table 1 there is “229” in the bottom right cell, not sure that is supposed to be there.

The number is not visible in the .docx file. It seems to be added during the PDF conversion

Rev. 2: Supplemental Figure 7 is too small to read the values and look at the shells, please put these on multiple pages and enlarge images figure Fig. 5. Same for line 254. Make the scale for El/Ca for each shell the same so the reader can compare visually. These figures should show microgranular and fibrous data.

We updated the figure according to your suggestions. 

Rev. 2: Table 4 How many annual cycles or years were compared for these shells, i.e. are the authors comparing the same time intervals or varying time intervals for all the shells. Looking a Fig S7, the data spans different number of years in the shells. The same interval and same amount of data of time should be made for the comparison tests.

The data compared here consists of different time interval for each sample. However, we only compare the data between the microgranular and fibrous shell section which spans the same interval for each sample.

Rev. 2:Table 5 Explain in the caption. Explain in the caption how All samples was determined, the same time intervals, or just all the data produced? If different intervals, then sample size needs to be considered.

For the correlation with temperature and salinity we used the same time interval. The values given under all samples relate to a combined regression with all samples in the same time interval. 

Rev. 2:Table 6 Explain in results or in the methods how the multiple linear regression was performed, give citation or software used.

We added a section about the statistical analysis, which program we used and the used tests.

Rev.2: Line 389 Spelling of “therefor” is incorrect.

Corrected. L397

Rev. 2:Line 398 What is the spatial-temporal resolution of the authors’ study? Add this to the table with the time interval present in the data for each shell that I requested.

For the statistical analysis we interpolated the data to a daily resolution.

Rev. 2:excavata are not well suited for paleo temperature and salinity reconstructions due to weak correlations between proxy trace elements and these environmental variables,…” Same for Line 547

Revised. “…section indicate that elemental ratios in A. excavata are not well suited for paleo reconstructions due to weak correlations between proxy trace elements and the environmental variables temperature and salinity…”

Rev. 2:Conclusions- I like the discussion and conclusions- use these to revise the abstract.

We revised the abstract.

Reviewer #3: The authors addressed my concerns from the previous revisions. The data and the study will be useful for other researchers and now that the others have more carefully assessed the physiological/vital effects that obscure the recorded environmental signals in the shell elemental ratios.

We want to thank the reviewer for the comprehensive comments which drastically improved this manuscript.

Reviewer #4: I grateful to editor the opportunity to review this manuscript. After reading it, unfortunately I should recommend its rejection. They have done a very good job and the methodology seems very strong. However, I am not very confident in the value of this manuscript for a very high-quality journal like Plos One.

The study conducted by the authors seems based on a very strong-methodology, but the results obtained have not implications for future climate studies. Please, take into account that it is not my subjective opinion, authors are who conclude the manuscript writing: "it can be stated that Mg/Ca, Sr/Ca and Na/Ca ratios in A. excavata are unlikely to be good proxies for environmental reconstructions at this point". They also admit it in the discussion section: “…these results indicate that Mg/Ca, Sr/Ca and Na/Ca are unlikely to be used for environmental reconstruction”. I agree, these E/Ca ratios are not reliable climate proxies. In four shells, they found a correlation lower than 0.1 between SST and Mg/Ca ratios. Mean correlations between E/Ca ratios and SST or Salinity are 0.06, 0.02, 0.17, 0.02, 0.04 and 0.12. I am so sorry, but these coefficients are not reporting a weak correlation as authors said, they are actually showing an absent of correlation. In this same sense, the authors are who admit that only 6% of the variability of Mg/Ca ratios can be explained by SST changes through mollusk life span. It is not possible to say at same time that 0.06 is a weak correlation while then they say that magnesium precipitation is only driven by SST in a 6%. In summary, manuscript have valuable data, a strong methodology, but the results, conclusions and specially its implications for future studies are not good enough for a very high impact journal like this.

In any case, if editor decide to admit it or if the other reviewers have favorable opinions, please check the manuscript again very carefully, there are several mistakes. Some examples: L. 155: Fig. 1 should be Fig. 2. L. 287. You said that maximum correlation value is 0.64, but in the table 5 the highest value is 0.63. L. 389. Please, correctly write “therefore”.

We adjusted the wording to be more careful with the reported results and corrected your mentioned errors

References

1. Gabitov RI, Sadekov A, Leinweber A. Crystal growth rate effect on Mg/Ca and Sr/Ca partitioning between calcite and fluid: An in situ approach. Chem Geol. 2014;367: 70–82. doi:10.1016/j.chemgeo.2013.12.019

2. López Correa M, Freiwald A, Hall-Spencer J, Taviani M. Distribution and habitats of Acesta excavata (Bivalvia: Limidae) with new data on its shell ultrastructure. Cold-Water Corals Ecosyst. 2005; 173–205. doi:10.1007/3-540-27673-4_9

3. Stainbank S, Kroon D, Rüggeberg A, Raddatz J, De Leau ES, Zhang M, et al. Controls on planktonic foraminifera apparent calcification depths for the northern equatorial Indian Ocean. PLoS One. 2019;14: e0222299. doi:10.1371/journal.pone.0222299

4. Higuchi T, Fujimura H, Yuyama I, Harii S, Agostini S, Oomori T. Biotic Control of Skeletal Growth by Scleractinian Corals in Aragonite–Calcite Seas. Roberts JM, editor. PLoS One. 2014;9: e91021. doi:10.1371/journal.pone.0091021

5. Shelswell KJ, Beatty JT. Coordinated, Long-Range, Solid Substrate Movement of the Purple Photosynthetic Bacterium Rhodobacter capsulatus. Otto M, editor. PLoS One. 2011;6: e19646. doi:10.1371/journal.pone.0019646

6. Bolz M, Ruggli N, Ruf M-T, Ricklin ME, Zimmer G, Pluschke G. Experimental Infection of the Pig with Mycobacterium ulcerans: A Novel Model for Studying the Pathogenesis of Buruli Ulcer Disease. Small PLC, editor. PLoS Negl Trop Dis. 2014;8: e2968. doi:10.1371/journal.pntd.0002968

7. Klishko OK, Lopes-Lima M, Bogan AE, Matafonov D V., Froufe E. Morphological and molecular analyses of Anodontinae species (Bivalvia, Unionidae) of Lake Baikal and Transbaikalia. Vermeij GJ, editor. PLoS One. 2018;13: e0194944. doi:10.1371/journal.pone.0194944

8. Yates AM, Neumann FH, Hancox PJ. The Earliest Post-Paleozoic Freshwater Bivalves Preserved in Coprolites from the Karoo Basin, South Africa. Farke AA, editor. PLoS One. 2012;7: e30228. doi:10.1371/journal.pone.0030228

9. Purroy A, Šegvić-Bubić T, Holmes A, Bušelić I, Thébault J, Featherstone A, et al. Combined Use of Morphological and Molecular Tools to Resolve Species Mis-Identifications in the Bivalvia The Case of Glycymeris glycymeris and G. pilosa. Vermeij GJ, editor. PLoS One. 2016;11: e0162059. doi:10.1371/journal.pone.0162059

---

## [Decision Letter · Decision Letter 2]

5 Jan 2021

Compositional variability of Mg/Ca, Sr/Ca, and Na/Ca in the deep-sea bivalve *Acesta excavata* (Fabricius, 1779)

PONE-D-20-12353R2

Dear Dr. Schleinkofer,

We’re pleased to inform you that your manuscript has been judged scientifically suitable for publication and will be formally accepted for publication once it meets all outstanding technical requirements.

Kind regards,

Tim M. Conway, Ph.D.

Academic Editor

PLOS ONE

Additional Editor Comments (optional):

The revised manuscript has been seen again by Reviewer 2, who now recommend acceptance with no further changes, adding to Reviewer 3's recommendation of acceptance in the previous round. I note that Reviewer 4 did recommend reject, but this was based on their assessment of the implications of the paper rather than the methodological quality - indeed, they highlighted the high quality of the methods. Their finding of high quality science is certainly consistent with PLOS criteria of publishing high quality science over considerations of significance, and so I have not given weight to their concerns in coming to an editorial decision. I am therefore pleased to note that all minor revisions have been made and I am now delighted to recommend acceptance at this time.

Reviewers' comments:

Reviewer's Responses to Questions

**Comments to the Author**

1. If the authors have adequately addressed your comments raised in a previous round of review and you feel that this manuscript is now acceptable for publication, you may indicate that here to bypass the “Comments to the Author” section, enter your conflict of interest statement in the “Confidential to Editor” section, and submit your "Accept" recommendation.

Reviewer #2: All comments have been addressed

2. Is the manuscript technically sound, and do the data support the conclusions?

Reviewer #2: Yes

3. Has the statistical analysis been performed appropriately and rigorously? 

Reviewer #2: Yes

4. Have the authors made all data underlying the findings in their manuscript fully available?

Reviewer #2: Yes

5. Is the manuscript presented in an intelligible fashion and written in standard English?

Reviewer #2: Yes

6. Review Comments to the Author

Reviewer #2: The paper is much improved and only one small typo was notice in the figure captions (space missing between word and parenthesis. Thank you for your hard work on this paper.

Please add the link and DOI for the data submission to Pangaea to the Acknowledgements or in the reference section for the final publication..

7. PLOS authors have the option to publish the peer review history of their article (what does this mean?). If published, this will include your full peer review and any attached files.

Reviewer #2: No

---

## [Editor Report · Acceptance letter]

21 Apr 2021

PONE-D-20-12353R2 

Compositional variability of Mg/Ca, Sr/Ca, and Na/Ca in the deep-sea bivalve *Acesta excavata* (Fabricius, 1779) 

Dear Dr. Schleinkofer:

I'm pleased to inform you that your manuscript has been deemed suitable for publication in PLOS ONE. Congratulations! Your manuscript is now with our production department. 

Kind regards, 

on behalf of

Dr. Tim M. Conway 

Academic Editor

PLOS ONE